# Developmental nonlinearity drives phenotypic robustness

Rebecca M. Green [1], Jennifer L. Fish [2], Nathan M. Young [3], Francis J. Smith[4], Benjamin Roberts [2], Katie Dolan[2], Irene Choi[4], Courtney L. Leach[1], Paul Gordon[5], James M. Cheverud[6], Charles C. Roseman[7], Trevor J. Williams[4], Ralph S. Marcucio [3] & Benedikt Hallgrímsson [1]

Robustness to perturbation is a fundamental feature of complex organisms. Mutations are the raw material for evolution, yet robustness to their effects is required for species survival. The mechanisms that produce robustness are poorly understood. Nonlinearities are a ubiquitous feature of development that may link variation in development to phenotypic robustness. Here, we manipulate the gene dosage of a signaling molecule, *Fgf8*, a critical regulator of vertebrate development. We demonstrate that variation in *Fgf8* expression has a nonlinear relationship to phenotypic variation, predicting levels of robustness among genotypes. Differences in robustness are not due to gene expression variance or dysregulation, but emerge from the nonlinearity of the genotype–phenotype curve. In this instance, embedded features of development explain robustness differences. How such features vary in natural populations and relate to genetic variation are key questions for unraveling the origin and evolvability of this feature of organismal development.

[1] Department of Cell Biology & Anatomy, Alberta Children's Hospital Research Institute and McCaig Bone and Joint Institute, Cumming School of Medicine, University of Calgary, Calgary, AB T2N 4N1, Canada. [2] Department of Biological Sciences, University of Massachusetts Lowell, Lowell, MA 01854, USA. [3] Department of Orthopaedic Surgery, School of Medicine, University of California San Francisco, San Francisco, CA 94110, USA. [4] Department of Craniofacial Biology, School of Dental Medicine, University of Colorado Anschutz Medical Campus, Aurora, CO 80045, USA. [5] Alberta Children's Hospital Research Institute, Cumming School of Medicine, University of Calgary, Calgary, AB T2N 4N1, Canada. [6] Department of Biology, Loyola University Chicago, Chicago, IL 60660, USA. [7] Department of Animal Biology, University of Illinois Urbana Champaign, Urbana, IL 61801, USA. Rebecca M. Green and Jennifer L. Fish contributed equally to this work. Ralph S. Marcucio and Benedikt Hallgrimsson jointly supervised this work. Correspondence and requests for materials should be addressed to R.S.M. (email: Ralph.Marcucio@UCSF.edu) or to B.H. (email: bhallgri@ucalgary.ca)

Waddington proposed that selection tends to stabilize development along particular paths, a phenomenon he called "canalization"[1]. He tested this idea by selecting for an induced trait in the presence of a teratogen (e.g., ether and the bithorax phenotype) and obtained individuals in which the trait appeared without the teratogen[2]. He hypothesized that selection had stabilized development around the induced trait such that it no longer needed the environmental stimulus. Concurrent work by Waddington and others showed that mutations with major effects tended to be more variable than the wild type[3–6]. This observation was also explained by invoking canalization. Mutations were hypothesized to increase variance by disrupting evolved mechanisms that buffered variation around a phenotypic mean[7]. This tendency for resistance to perturbation in development, or robustness, is widely thought to be a fundamental property of complex life[8]. Yet, the mechanisms responsible for promoting and modulating robustness remain largely unknown[9].

Wagner et al.[10] defined canalization as suppression of phenotypic variation among individuals due to insensitivity to either genetic or environmental effects. This definition hinges on a distinction between the frequency distribution of the genetic or environmental factors that cause variation and the magnitudes of phenotypic effect associated with those factors. A mutation or environmental effect disrupts or decreases canalization when phenotypic variance is increased, while all other genetic or environmental effects are unchanged.

Two kinds of mechanisms have been proposed to explain canalization. In one, specific molecular mechanisms such as heat shock and other chaperone proteins[11–14] or microRNAs[15] buffer against perturbations and suppress the expression of variation. In the other, canalization emerges from redundancies, feedback loops, and other features of developmental systems[9,16–20]. These explanations are not mutually exclusive,

and multiple mechanisms may act simultaneously at different levels of development[9]. However, they differ in that one posits the existence of organism-wide buffering processes that reduce variation, while the other holds that robustness emerges from the same mechanisms that generate variation in specific traits. A common feature of developmental systems explanations for robustness is the importance of nonlinearity[21–24]. Ligand–receptor binding, often described with a Hill function, is commonly nonlinear[25]. The same is true for transcriptional regulation[26]. Within tissues, processes such as the diffusion of a morphogen are nonlinear in ways that depend on anatomical context[27]. Genetic variation influences the phenotype via developmental processes that act at different scales, times, and locations within the organism, complicating the relationship between genotype and phenotype[17,28]. Therefore, it is not clear how nonlinearities in specific mechanisms translate to quantitative relationships between genetic and phenotypic variation.

Lewontin introduced the genotype–phenotype (G–P) map to conceptualize relationships between genetic and phenotypic variation[29]. G–P maps are often nonlinear, as evident in dominance and epistasis[30,31]. While much has been learned about the developmental mechanisms that construct vertebrate morphology, much less is known about the relationship between developmental and quantitative phenotypic variation. Alberch[32] suggested a framework for incorporating development into G–P maps, and Rice[33] developed quantitative genetic theory to formally relate variation in development to phenotypic variation. Curvatures in the developmental landscape indicate nonlinear relationships between developmental processes and phenotypic variation. More recently, Morrissey[34] provided a theoretical framework to quantitatively relate developmental and phenotypic variation for such nonlinearities. A consequence of such nonlinear G–P relations is modulation of the amount of phenotypic

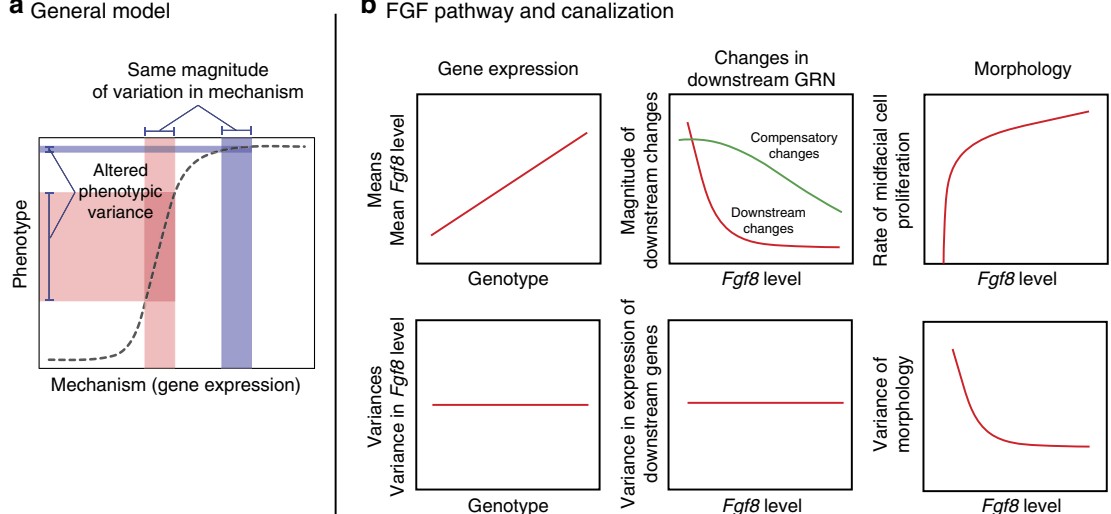

**Fig. 1** Nonlinearities at multiple levels across development modulate variance. **a** General model of a nonlinear genotype–phenotype map where the amount of a particular developmental process (e.g., cell survival, proliferation, and Fgf signaling) determines mean phenotype. Note that the same amount of variation in the mechanism ("wild-type" gene expression—blue vertical bar, "mutant" gene expression—red vertical bar) can generate vastly different amounts of phenotypic variation. This model yields a canalized region where variance is buffered ("wild-type" shape variation, blue horizontal bar) and an area where canalization is lost ("mutant" shape variation, red horizontal bar). **b** Hypothetical model of how nonlinear genotype–phenotype relationships are generated at multiple biological levels. The top left panel shows that gene expression will relate linearly to cranial phenotype. The top mid panel shows that changes in the gene regulatory network (GRN) downstream to *Fgf8* respond either nonlinearly, driving change in the phenotypic mean (red line), or act in a compensatory manner, buffering the effect of variation in *Fgf8* (green line). The top right panel shows that morphology will relate nonlinearly to Fgf8 level, potentially due to nonlinear changes in the underlying cell biological processes. Variances are influenced at the level at which the nonlinearity arises (lower panels)

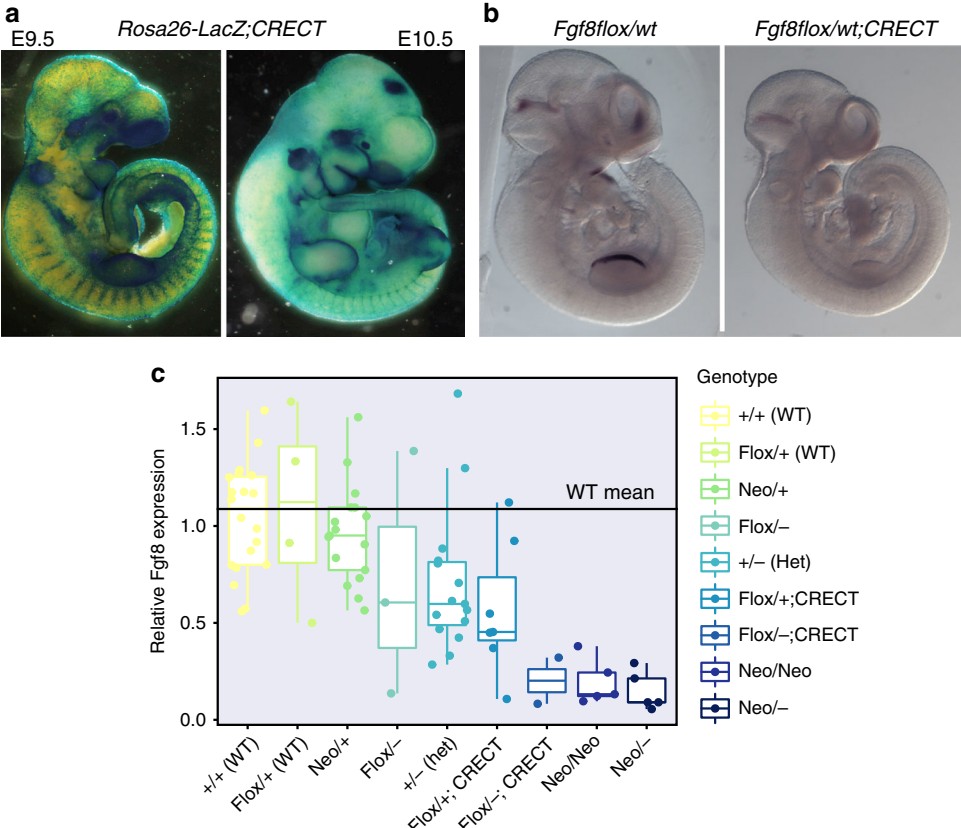

**Fig. 2** Generation of the allelic series. **a** E9.5 and E10.5 expression of CRECT as detected by crossing CRECT males with B6;129S4-*Gt(ROSA)26Sortm1Sor*/J (R26R) females and staining the embryos for beta-galactosidase expression. Note the thin layer of blue present over the entire embryo showing the ectodermal CRE expression. **b** In situ hybridization showing regions of decreased *Fgf8* in the E10.0 *Fgf8flox/flox;Crect* embryos. **c** qRT-PCR of cranial tissue showing *Fgf8* levels by genotype; sample size is between 2 and 22 samples per group. The box represents 1.5× the interquartile range of the data. Allelic series for *Fgf8* generates gradual loss of *Fgf8* mRNA. Data shown is the delta–delta-CT value, where data were normalized against the mean delta-CT for the WT group. The homozygous null is not included as it is lethal

variance for a given amount of variation in some developmental factor (Fig. 1a)[16–18,35].

We previously demonstrated significant nonlinearity in the relationship between sonic hedgehog signaling and embryonic facial shape[36]. Variation in the three-dimensional morphology of the face is far removed from nonlinear molecular processes or the theoretical dynamics of gene regulatory networks. For this reason, it is not at all clear that the theoretical predictions that link nonlinearity to phenotypic variance should hold across the vast complexity of the G–P map. To test the hypothesis that a nonlinear G–P relationship predicts variation in robustness, we examine how variation in *Fgf8* expression affects the mean and phenotypic variance for craniofacial shape.

*Fgf8* is appropriate for this study as it drives a central pathway in craniofacial development[37–39]. *Fgf8* is a signaling factor that is expressed in the facial and oral ectoderm, where it directs craniofacial pattern and polarity[40,41]. *Fgf8* is absolutely required for proper development of facial structures[42,43]. *Fgf8*-expressing cells form a boundary with *Shh*-expressing cells to form the frontonasal ectodermal zone, which directs the outgrowth of the facial prominences and has also been implicated in their evolution[44,45].

We predict that *Fgf8* expression relates nonlinearly to craniofacial phenotype. Further, we predict that the shape of the curve relating mean phenotype to *Fgf8* level will dictate the phenotypic variance within and between genotypes. Genotypes falling on the steeper portions of the curve will have higher variances (differences among individuals within genotype) than

the genotypes falling on flatter portions. Likewise, different genotypes that fall along the steeper portions of the curve will have higher genetic variances, while those along the flatter portion of the curve will show little phenotypic variation (Fig. 1a). At the transcriptome level, we further predict that there will be both compensatory and downstream gene expression changes (Fig. 1b). We show that once *Fgf8* falls below a threshold level, there is both a change in the mean cranial shape and an increase in the variance of that shape. We further show that changes in phenotypic variance do not relate to increases in gene expression variance and that there are both nonlinear and linear downstream gene expression changes.

## Results

**Allelic series generation.** To modulate *Fgf8* expression during facial morphogenesis, we used two allelic series of mice varying in *Fgf8* dosage. The first, Fgf8neo, was generated from the *Fgf8* neomycin cassette insertion series[46]. This series includes a full null allele, as well as a hypomorphic allele due to the retention of the neomycin insertion. The second series, *Fgf8;Crect* uses the floxed allele that was generated after the removal of the neomycin cassette to delete *Fgf8* specifically in the ectoderm using the ectodermal cre, *Crect*[47] (Fig. 2a). In the Fgf8neo series, *Fgf8* levels are affected globally from fertilization[46]. *Fgf8;Crect* embryos show loss of *Fgf8* in the ectoderm and decreased *Fgf8* in the forebrain beginning by E10.0 (Fig. 2b). We chose these two series because their combination results in nine alleles of *Fgf8* generating a series

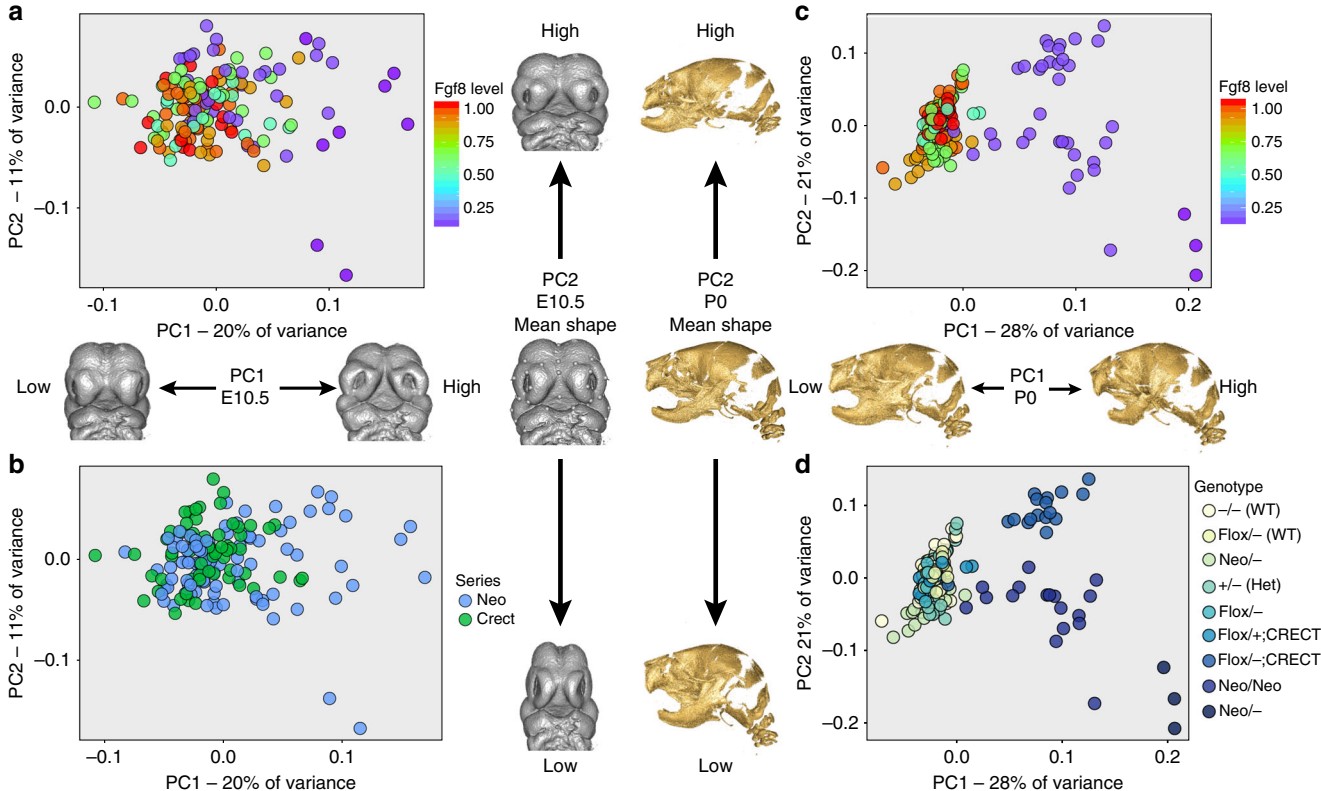

**Fig. 3** Shape changes in response to decreased *Fgf8* gene dosage. Principal component analysis (PCA) of shape at E10.5 (**a**, **b**) and P0 (**c**, **d**). Gray embryos (E10.5) show shape change trajectories for PC1 (horizontal) and PC2 (vertical), and the middle vertical image represents the mean shape for each time point. Gold skulls show the same shape change trajectories for the P0 data. **a**, **c** PC plots colored by *Fgf8* level with warm colors representing wild-type embryos and cool colors and purples showing around 20% *Fgf8* mRNA expression (mean per group by qRT-PCR). **b** Coloration by genotypic series, genotypes separate by allelic series, but the differences between low PC1 and low PC2 are small. The mean shape of individual genotypes is already significantly different (Procrustes permutation test, *P* <0.001) by E10.5. **d** Coloration by genotype as used in the rest of the paper. A total of 187 neonates were analyzed and divided between groups as follows: WT (+/+) = 22, Flox/+ = 29, Neo/+ = 41, Flox/− = 10, ± = 25, Flox/+;Crect = 21, Flox/−;Crect = 19, Neo/Neo = 17, and Neo/− = 3 (w/all landmarks present). A total of 156 embryos were analyzed and divided between groups as follows: WT (+/+) = 27, Flox/+ = 15, Neo/+ = 30, Flox/− = 13, ± = 16, Flox/+;Crect = 16, Flox/−;Crect = 19, Neo/Neo = 12, and Neo/− = 8

of gradations in *Fgf8* dosage (Fig. 2c, Supplementary Table 1). Mean *Fgf8* levels in the head for the nine genotypes relative to the wild-type embryos from the *Fgf8*neo series vary significantly (ANOVA, df = 69, 8, $P < 1*10^{-7}$), ranging from 0.14 to 1.1 (Fig. 2c), yet we detect no difference in the variance of gene expression across the genotypes (Levene's test, df = 69, 8, $P = 0.2043$). Further, by deleting *Fgf8* in two different ways, we are able to show consistency between different mechanisms of *Fgf8* loss. Facial phenotype is assessed by geometric morphometric analysis[48,49] at embryonic day 10.5 (E10.5) and immediately after birth, postnatal day 0 (P0). These time points capture early face formation and late fetal skull formation.

**Generation of a genotype–phenotype map**. To determine the shape of the G–P map for *Fgf8* expression, we determined *Fgf8* expression by quantitative real-time PCR (qRT-PCR) of the head and craniofacial shape via three-dimensional landmark-based geometric morphometrics[48,50]. Here, the perturbation is the modification of *Fgf8* level across genotypes, while the phenotype is a multivariate measure of facial shape as determined from three-dimensional landmark data. The nine genotypes also vary significantly in facial shape at both E10.5 and P0, as determined by ANOVA (*P* <0.01). Using principal component analysis (PCA), we determined that the allelic series ordinates along the first principal component (PC) of craniofacial shape (Fig. 3). At both developmental stages, *Fgf8* expression accounts for a

significant proportion of shape variation (7.1% of shape variation at E10.5 and 16.4% at P0), as determined by multivariate regression after standardizing for embryo age (E10.5) or size (P0)[51]. At E10.5, the genotypes vary along PC1 by *Fgf8* level (Fig. 3a, b). A similar pattern is seen at P0, showing that the correlation is preserved throughout embryogenesis (Fig. 3c, d).

To model the relationship between *Fgf8* expression and phenotypic variation, we used Morissey's[34] quantitative model for nonlinear G–P maps. This model produces a prediction of the amount of variance that should be observed given a nonlinear G–P map. To generate the curve used to test Morissey's model, we fit the *Fgf8* gene expression data, and the phenotypic data (3D landmark data) to a von Bertalanffy curve using least-squares regression. The phenotype data used was the regression score from a multivariate regression of our normalized Procrustes coordinates on *Fgf8* level—which generates single variable shape score[48]. These curves are shown in Fig. 4a, b.

**Loss of *Fgf8* affects phenotypic variance**. The Morissey model, based on the mean and standard deviation of our *Fgf8* gene expression data, predicts that variation in *Fgf8* expression has little effect on shape metrics (phenotypic value or regression score) when *Fgf8* expression is >40% of the wild-type level, while below this point, variation in *Fgf8* expression produces increasingly large effects on the mean phenotype. Figure 4c, d shows the predicted relationship between the variance of *Fgf8*

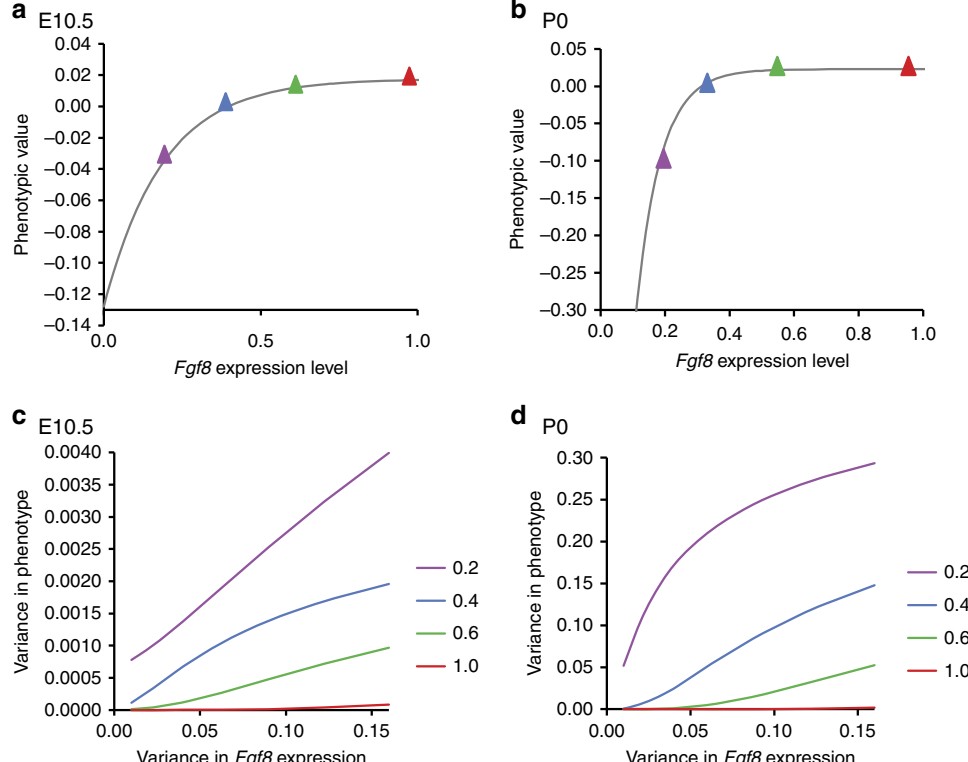

**Fig. 4** Mathematical modeling of phenotypic variance. **a, b** Fitting the shape data (regression residuals) to a nonlinear, von Bertalanffy growth curve. Colored dots highlight the location on curves of four gene expression values that are modeled in **b, c**. Least-squares regression models the curve as $z = 0.01765 - (0.01765 - (-0.12787))e^{(-5.3003 \cdot x)}$ at E10.5 (A) and $z = 0.0288 - (0.0288 - (-1.333))e^{(-13.049 \cdot x)}$ at P0. **c, d** Predicted relationship between the variance of $Fgf8$ expression and phenotypic variance at four different levels of $Fgf8$ expression. Expression levels were not extrapolated below zero. This led to the use of truncated normal distributions for expression variance and is responsible for the nonlinearities in **c, d**

expression and phenotypic variance at four mean expression levels for the E10.5 and P0 samples. These results show that the variance of $Fgf8$ expression will have little effect on phenotypic variance when $Fgf8$ level is >50% of the wild-type level, while the phenotypic variance becomes increasingly sensitive to gene expression variance below this point.

Figure 5a, b shows the individual-level data for the regression of shape against mean $Fgf8$ level. The von Bertalanfy curve explains 54% of the phenotypic variance at E10.5 and 84% at P0. $Fgf8$ expression measured by RT-PCR in the head relates nonlinearly to craniofacial morphology. Following the prediction of the Morrissey model, when $Fgf8$ levels are above 40% of wild-type levels, the effect on mean shape is minimal. Below this point, however, the phenotype deviates sharply. When $Fgf8$ expression levels are reduced below 40% of wild-type levels, small differences in $Fgf8$ expression have large phenotypic effects.

To determine whether nonlinearity predicts robustness, we plotted variance in face shape against $Fgf8$ expression across genotypes. No change in shape variance, measured as the Procrustes variance or morphological disparity[48,49], is seen until $Fgf8$ expression drops below 40% of wild-type levels (Fig. 5c). As predicted, shape variance dramatically increases below 40% expression in both the E10.5 and P0 samples, corresponding to the point at which the phenotype becomes sensitive to $Fgf8$ levels (P values between groups—Supplementary Table 2). The only exception is for E10.5 $Fgf8$;$Crect$ embryos, likely due to the fact that Crect does not activate until E9.5. By P0, this group has significantly increased phenotypic variance (Supplementary Table 2). At P0, the $Fgf8neo/-$ embryos are so highly dysmorphic that most of them could not be landmarked and so were not included in the variance analysis (Supplementary Fig. 1).

**Effects on gene expression**. To eliminate the possibility that differences in genetic variance across the allelic series account for the differences in phenotypic variance, we quantified genetic variance from high-resolution SNP data. These results show no relationship between genetic variance and phenotypic variance across the allelic series (Supplementary Fig. 2).

We determined the genome-wide changes in expression across the allelic series at E10.5 using RNAseq. We reduced the transcriptome data using PCA. The pattern of gene expression within genotypes varies across the allelic series. PC1 of the transcriptome accounts for 44% of variation in gene expression. We interpret this PC to reflect the coordinated genome-wide changes in gene expression across the allelic series. This PC1 ordinates the allelic series (Fig. 6a). Mean $Fgf8$ expression level by genotype accounts for 30% of the variation in PC1 of the transcriptome (Fig. 6b). For further analysis, the data were separated into three groups: all genes, the MapK Kegg pathway, and a hand-curated list of 15 known, direct Fgf target genes (Supplementary Table 3). The MapK Kegg pathway was selected as Fgf signaling falls within the MapK signaling cascade. In all the genes and in the MapK pathway, there appears to be a curve in the data; however, each group is statistically different from its neighbor (Fig. 6c, d). This nonlinearity becomes more pronounced for the Fgf downstream targets. For these genes, expression level does not differ significantly among the heterozygote genotypes (T test, $P = 0.96$; Fig. 6e). The lack of change between these groups generates a flat region in the curve with an inflection point at 40–50% of the wild-type $Fgf8$ level, demonstrating nonlinearity.

To test the hypothesis that the low $Fgf8$ expression genotypes have increased phenotypic variation because of less coordinated

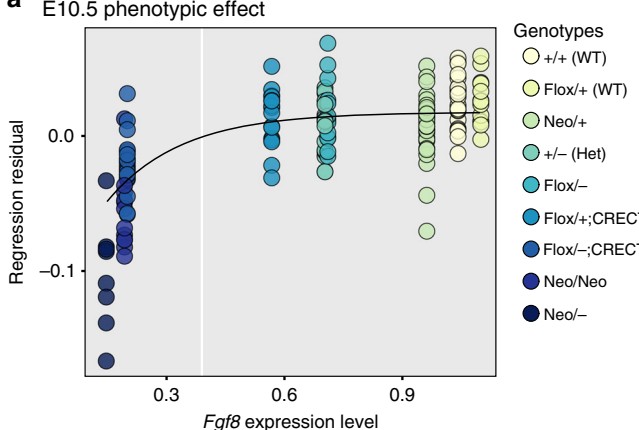

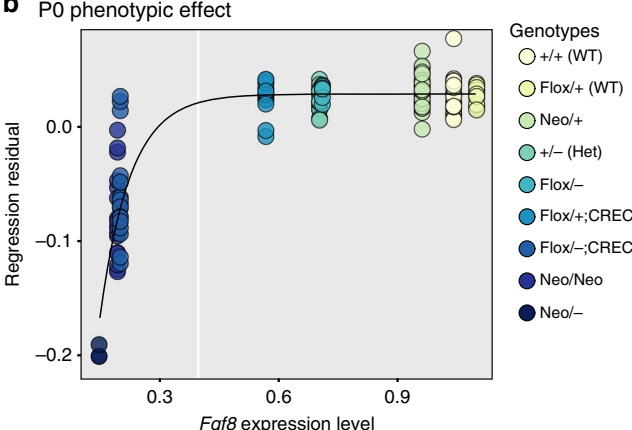

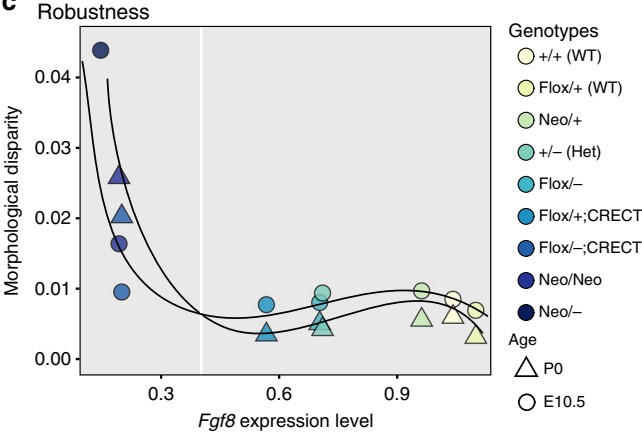

**Fig. 5** Shape and shape variance relate nonlinearly to *Fgf8* mRNA expression. Shape is defined using the common allometric component of shape (CAC). **a**, **b** Multivariate regression of shape on *Fgf8* level at **a** E10.5 and **b** P0. The black line shows the von Bertalanffy curve modeled in Fig. 4. **c** Variance as calculated by the Procrustes variance or morphological disparity[49,80]. The white vertical line shows an apparent threshold near 40% of wild-type *Fgf8* level. *P* values between groups are shown in Supplementary Table 2

among individuals. High correlations indicate a high degree of consistency among individuals within genotypes. We performed this analysis for both genome-wide and for each of the two groups of genes known to be downstream of *Fgf8*. This analysis revealed no evidence of dysregulation of gene expression across the allelic series. The pairwise correlations genome wide or within likely downstream targets show no detectable pattern across genotypes (Fig. 6f–h).

To determine whether the transcriptomic data show evidence of compensatory changes that could explain the lack of phenotypic response above 40% of the wild-type *Fgf8* expression level, we searched for significant correlations between groups of genes and *Fgf8* level across individuals and genotypes. Resampling revealed elevated ($P < 0.05$) correlations for the reactome Fgf downstream-signaling pathway, but not for Wnt, apoptosis, MapK Kegg, and hedgehog pathways. Eight individual genes fell outside of the 95% confidence interval based on genome-wide resampling of a similar number of genes. This list includes *Fgf4* and *Trib3* that are negatively correlated with *Fgf8*, and *Fgf17*, *Etv4*, *Prkcg*, *Spry1*, *Spry4*, and *Rictor* that correlate positively. The two Sprouty (Spry) genes and *Etv4* are known to be downstream of *Fgf8*, suggesting that *Fgf8* signaling modulates downstream genes across the entire range of expression. A MANOVA shows that the genotypes vary significantly in the expression of these downstream genes (Pillai's trace = 2.1, $P < 1*10^{-5}$). As a confirmation, we performed RT-PCR analysis on these eight genes and compared them against the *Fgf8* levels for each sample (Fig. 7). While *Fgf4* failed to reach statistical significance, *Trib3* does appear to be weakly, but significantly negatively correlated with *Fgf8* level. All other genes were positively and significantly correlated with *Fgf8* level. *Fgf17*, for example, trends toward mild upregulation in the Neo/+ group ($1.27 \pm 0.16$ vs. $1.06 \pm 0.32$, Student's *t* test, $P = 0.15$). Mean levels by genotype and standard deviations have been listed in Supplementary Table 4. These results suggest that there may be genes that change in expression to compensate for loss of *Fgf8*, though this requires further investigation.

## Discussion

We show that nonlinearity in the G–P relationship for *Fgf8* expression predicts phenotypic robustness. Progressive reduction in *Fgf8* yields a nonlinear relationship to phenotype, affecting both mean facial shape and the magnitude of phenotypic variance. Development tolerates a large amount of change in *Fgf8* expression around wild type, but only to a point, after which small changes in *Fgf8* lead to large changes in phenotype, thus permitting more morphological variance to be generated in a population for a given amount of variation in *Fgf8*. These findings show that nonlinearity in a single pathway can propagate across the many levels of organization (molecular, cellular, tissue, etc.) that channel information from genotype to phenotype, providing a viable mechanistic explanation for canalization (Fig. 1).

Our results are consistent with the hypothesis that the nonlinear G–P map for *Fgf8* explains the differences in phenotypic variance across the *Fgf8* allelic series. There are minor differences among individuals in *Fgf8* expression within each genotype for the allelic series. Our model predicts that these minor differences will translate to different magnitudes of phenotypic variance along the curve that describes the relationship between *Fgf8* expression and the mean cranial phenotype at each point along the curve. This result implies that robustness can emerge in developmental context as a consequence of nonlinearities in development. This contrasts with explanations for canalization that involve dedicated mechanisms such as heat shock proteins that regulate variability organism wide. Our results do not

or dysregulated gene expression, we obtained the complete set of pairwise correlations between gene expression levels across individuals within genotypes. If phenotypic variances are low within a genotype, one might expect the genomes of individuals to be expressed in similar ways, while high phenotypic variance might be associated with large differences in gene expression

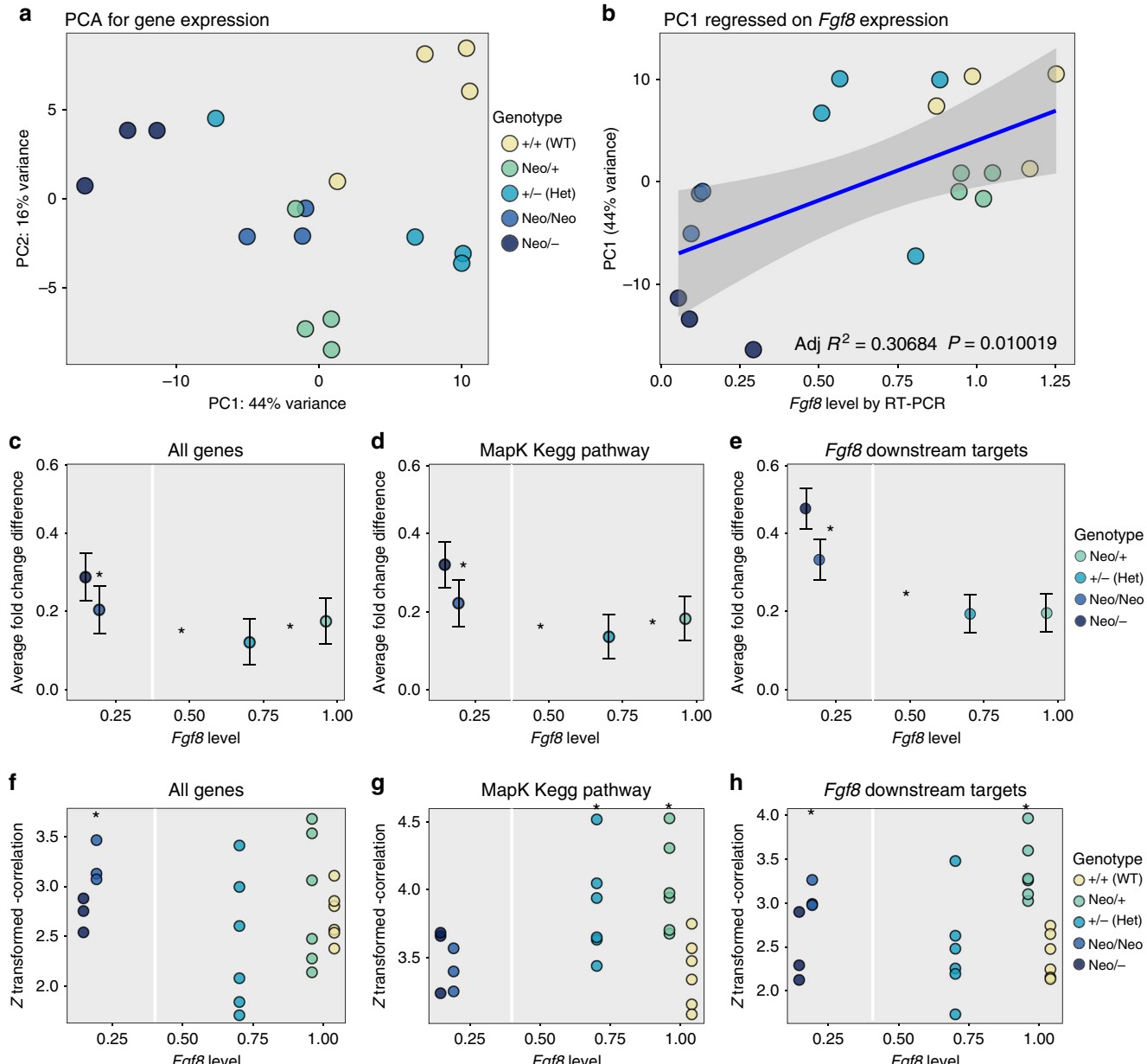

**Fig. 6** Gene expression changes across the *Fgf8* allelic series. **a** PC 1 and 2 plot of RNAseq data (18 samples). No differences in dispersion are observed between groups. **b** Relationship between PC1 of the RNAseq data and *Fgf8* level for each sample as quantified by RT-PCR. The blue line shows the line of best fit, gray shows the 20% error around the line. **c–e** Average absolute value fold change (dot) and average absolute value standard error of the fold change (error bar) between each mutant genotype and wild type, **c** all measured genes, **d** MapK Kegg gene list (174 genes), and **e** *Fgf8* downstream targets (15 genes). The asterisk represents *P* <0.05 (bootstrap resampling) between nearest-neighbor groups (shown between the groups). The white vertical line shows an *Fgf8* level of ~40%. **f–h** Z-transformed covariance between each embryo within a genotype on **f** all genes, **g** MapK Kegg gene list (174 genes), and **h** 15 known *Fgf8* downstream targets (Supplementary Table 3). The asterisk represents *P* <0.05 (bootstrap resampling) between group and wild type

preclude the existence of such mechanisms, but they provide an additional and, perhaps, more general explanation for genetic and environmental influences on phenotypic robustness.

An alternative explanation to the changes in variance along the range of *Fgf8* expression is that disruptions to *Fgf8* expression dysregulate downstream gene regulatory networks, producing increased variance in gene expression that translates to increased phenotypic variance. Such disruptions might be specific to downstream targets of *Fgf8* or be more widespread. By this explanation, differences among individuals are greater at the lower range of *Fgf8* expression because these individuals also vary in expression of downstream genes. It predicts that as *Fgf8*

dosage falls below the threshold, the variance of downstream gene expression increases. This explanation relies on the idea that extreme changes in gene expression may have systemic destabilizing effects on development. This is implicit in the *Hsp90*[12] explanation for the source of robustness, as well as in several older explanations for canalization such as Lerner's genetic homeostasis model[52]. However, we found no evidence of increased variance of gene expression, suggesting that the increased phenotypic variance in genotypes producing low levels of *Fgf8* is not attributable to greater instability of the downstream gene regulatory network.

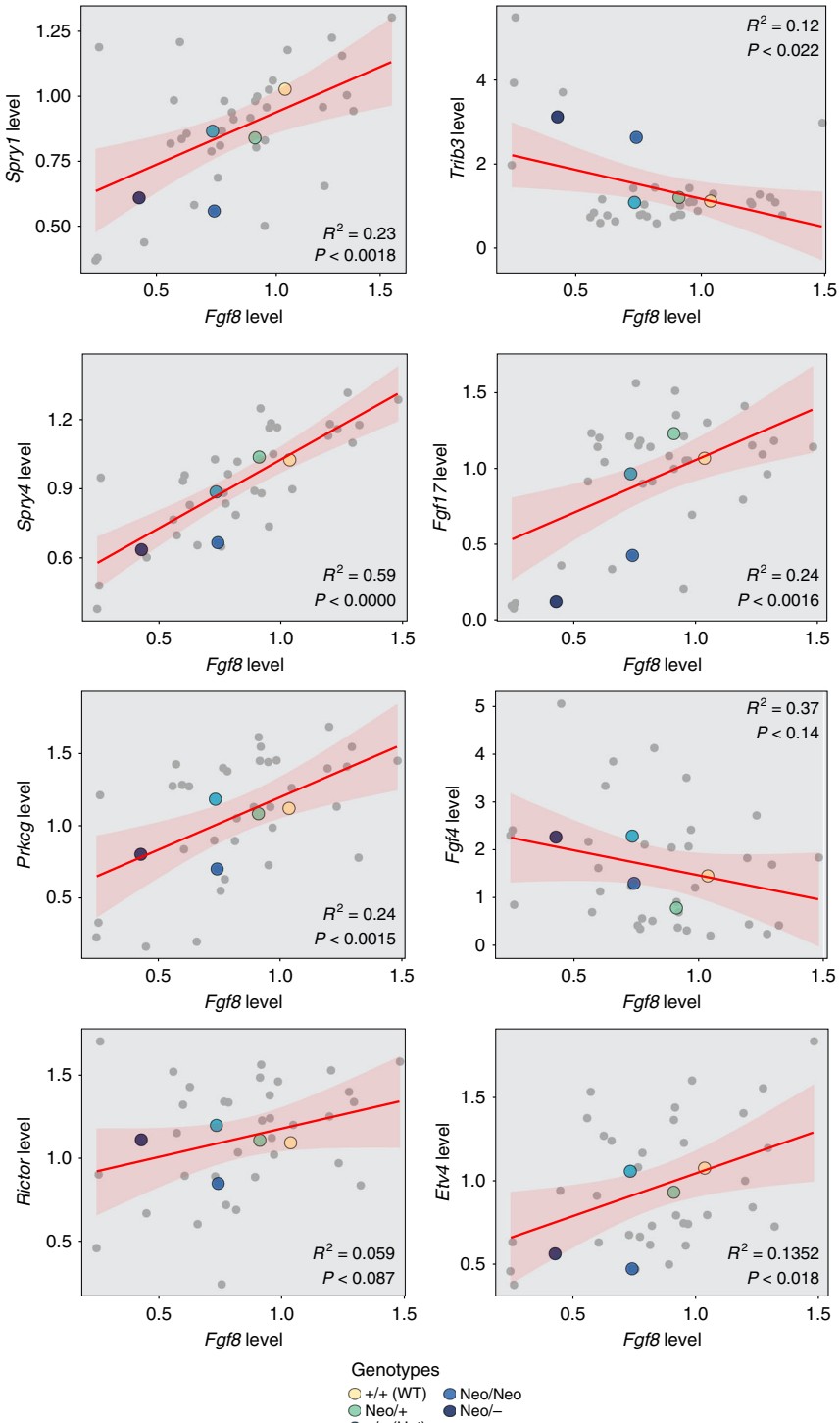

**Fig. 7** RT-PCR validation of correlation eight genes with *Fgf8* level. Thirty-seven samples from across the genotypes were analyzed for each of the eight genes plus *Fgf8* and modeled for a linear relationship. The linear relationship is shown in red (line±SE shaded). $R^2$ values and the adjusted *P* values from the linear model are shown

We did find, however, that genes downstream of *Fgf8* respond nonlinearly to *Fgf8* expression. In other words, the increased phenotypic effects at low *Fgf8* levels are mirrored by increased changes in gene expression, particularly in genes known to be downstream of *Fgf8*. This suggests that the nonlinear G–P map is a feature of a larger gene regulatory pathway, and that the phenotypic effects at low *Fgf8* levels are occurring because many genes are more responsive to *Fgf8* levels within that range than at levels closer to the wild type.

Interestingly, the phenotypic effects of the loss of *Fgf8* become more marked between E10.5 and P0. At P0, the genotypes separate more clearly and the increase in phenotypic variance at the steep part of the curve becomes more marked. *Fgf8* is expressed throughout facial prominence outgrowth and face formation[53]. Our results suggest that the effects of perturbing *Fgf8* expression below the threshold of 40% are exacerbated during late embryonic and fetal development.

Here, we build on earlier work in which we show a nonlinear G–P map for Shh expression and facial shape in chicks[36]. This study did not determine how phenotypic variance is modulated along the expression curve, however. Further, that study manipulated Shh expression directly rather than via a genetic model as we have done here. The advantage of the genetic approach is that we can eliminate experimental error as a source of among-individual variance within groups.

Our findings have important implications for the evolvability of morphology. Applying Morrisey's model[34] shows that even with strong selection on midfacial shape and substantial expression variation in Fgf8 levels, there would be little to no response to selection on facial morphology through alterations of Fgf8 expression levels. The correlated response of Fgf8 expression would be very low. On the other hand, at lower mean Fgf8 expression levels, response to selection on midfacial shape would be achieved, at least in part, by changes in Fgf8 expression. There would be a substantial correlated response in Fgf8 levels. These contrasting results flow directly from the nonlinear relationship between Fgf8 levels and midfacial morphology and suggest that while Fgf8 clearly plays a pivotal role in craniofacial development, it is unlikely to contribute directly to microevolutionary changes in craniofacial form under a wide range of expression levels, from 40% of wild-type expression to full wild-type expression.

Similarly, the nonlinear G–P map for Fgf8 expression and craniofacial shape helps us understand a puzzling and emerging trend in the genetics of complex traits. Why is it that the genes known from developmental biology to play major roles in the construction of morphology so often appear to play minimal roles in determining the variation of that morphology? Studies of craniofacial shape variation in mice and humans reveal a growing list of causal variants[54–56]. While some have known roles in facial development, many of the major players such as Shh or Fgf8 are conspicuously absent from these lists. Nonlinear G–P maps for such central genes would explain this result.

But how do nonlinear G–P maps for key developmental factors such as Fgf8 arise in the first place? The developmental origins of nonlinearities can be at various levels of organization from receptor ligand relationships to spatiotemporal tissue interactions. Simulations of developmental mechanisms such as Zhang et al.'s[57] multiscale model of limb development, often generate nonlinear effects simply as a consequence of spatiotemporal dynamics of cellular and tissue-level processes. Even so, nonlinear effects in development are presumably evolvable. For instance, the relationship between Fgf8 expression and its various downstream effects is likely heritable. Such nonlinearities might evolve through stabilizing selection acting on epistatic variance, although this has not been demonstrated in nature[8,58]. If this is true, then, genes deeply embedded within developmental systems, such as Fgf8 should relate more nonlinearly to phenotypic variation than genes with more peripheral roles. This might occur for key signaling factors like Fgf8 because insufficiency produces highly deleterious effects, while overexpression may have less deleterious consequences. Excess production of important proteins has been suggested as an explanation for canalization[59] and is also the basis for Sewall Wright's hypothesis for the developmental basis of dominance[60].

Canalization influences long-term evolvability because of an accumulation of cryptic variation that can be uncovered by changes in the genome or the environment[25]. Positing the existence of canalizing mechanisms that are specifically adapted to harbor reservoirs of variation requires an implausible group selection explanation. Our finding that nonlinearity in Fgf8 signaling modulates phenotypic robustness suggests instead that cryptic variation can emerge as a side effect of nonlinearities in developmental processes. Any genetic or environmental influence that affects a developmental factor that relates nonlinearly to a phenotype has the potential to affect the phenotypic variance[61]. Importantly, such genetic influences can just as plausibly be changes in allele frequencies as novel mutations.

A key challenge in evolutionary developmental biology is to relate the quantitative genetic theory that underpins evolutionary biology to developmental mechanisms. This is important because the evolvability of phenotypes is determined in large part by how development structures phenotypic variation[62–64]. Our study contributes to this goal by connecting the concept of canalization to developmental mechanisms. In quantitative genetics, gene interactions generate epistasis[65], and canalization can evolve by selection on epistatic variance[66]. However, once a nonlinearity occurs in development, it will generate gene interactions if the differential variation along the curve is heritable. Seen in this light, developmental nonlinearities are a cause rather than a consequence of epistasis. Epistasis is widely thought to contribute to missing heritability for complex traits because it can cause similarity among relatives not accounted for in QTL or GWAS studies[67]. For these reasons, the developmental basis for canalization is central to both the evolvability and the genetics of complex traits.

## Methods

**Mouse breeding and embryo generation.** The Fgf8neo series is a five-member series generated from a combination of the neomycin insertion into the intron between exons 2 and 3 of the Fgf8 locus and a null allele generated from loss of exon 2. The Fgf8;Crect series contains combinations of a floxed allele for Fgf8, a null allele for Fgf8, and then Fgf8 is deleted from the ectoderm around E9.5 using an ectodermal Cre (CRECT) (Fig. 2). The two series of mice were generated independently by different labs (Crect, T. Williams) (Neo, R. Marcucio/J. Fish). Both series of Fgf8 mice were generated from the Fgf8 flp/floxed allele originally developed by Meyers et al.[46]. The neo cassette was maintained in the Fgf8Neo mice. To generate the floxed allele for the CRECT studies, the neomycin resistance cassette was removed by crossing these mice to β-actin-flp (B6.CgTg (ACTFLPe) 9205Dym/J), generating the floxed allele. Deletion constructs were developed by crossing with β-actin Cre (FVB/N-Tg(ACTB-cre)2Mrt/J), to delete exons 2 and 3 from all cells.

To generate the Fgf8neo series, crosses were performed between mice that were heterozygous for the Neo (flp) allele or heterozygous for the Neo (flp) allele and the null allele. The Neo allele was genotyped with the following primers (5′–3′): F: CTG CAG AAC GCC AAG TA G; R: AGC TCC CGC TGG ATT CCT C. The null allele (UCSF/UMass) was genotyped with the following primers (5′–3′): F: GCC GTC TGA ATT TGA CCT GAG CGC; R: GAA ACC GAC ATC GCA GGC TTC TGC. The null and Neo alleles can be genotyped simultaneously at an annealing temperature of 58 °C. The floxed allele was genotyped using the following primers: (5′–3′) EM 99: CTT AGG GCT ATC AA CCC ATC and EM32: GGT CTC CAC AAT GAG CTT C. The null allele (UCDenver) was genotyped using EM41: AGC TCC CGC TGG ATT CCT C and EM99. These three can also be genotyped simultaneously at 58 °C. The Crect deletion series was generated by crossing the Crect, early ectodermal Cre (Fig. 2)[47], with the null allele, and then males from this cross were crossed to Fgf8 flox/flox females on an FVB background. Genotyping for this allele was performed using general Cre primers[68] (5′–3′): Cre1: GCT GGT TAG CAC CGC AGG TGT AGA G; Cre3: CGC CAT CTT CCA GCA GGC GCA CC with a 67 °C annealing temperature.

For embryos, pregnant dams were sacrificed at embryonic day (E) 10.5 based on visualization of a postcoital plug at E0.5. Embryos were dissected on ice and fixed in 4% paraformaldehyde and 5% glutaraldehyde prior to μCT scanning. Neonates were killed in $CO_2$ on ice and then fixed in 4% paraformaldehyde and 5% glutaraldehyde prior to μCT scanning.

Mouse experiments were approved by the UC Denver Institutional Animal Care and Use Committee (Crect series mice) and by the UCSF and University of Massachusetts Lowell Institutional Animal Care and Use Committee (Neo series mice).

The strains in the allelic series are highly inbred but not completely isogenic. We estimated genetic variation in each strain to verify that differences in phenotypic variance among genotypes are not explained by genomic variation (Supplementary Fig. 2).

**SNP analysis to estimate genetic variance.** Following genotyping, five DNA samples per each wild-type (WT), Neo/+, WT/−, Neo/Neo, and Neo/− groups were sent to GeneSeek Inc. (a NeoGene company, Lincoln, Nebraska USA). DNA samples were run on the GigaMuga mouse genotyping chips (Illumina), for a total of 143,000 SNPs. After quality control and removal of the X and Y chromosomes, we performed analyses using 133,559 SNPs on each of 17 samples, 4–5 per group. QC and SNP calls were done using the GenomeStudio Package (Illumina) by

GeneSeek. Further analysis was performed using the SNPRelate Package in R to calculate the SNP frequencies and the relative inbreeding[69]. The SNP frequencies were used to calculate the additive genetic variance[70].

**Scanning and landmarking**. All samples were µCT scanned on a µCT35 scanner to visualize facial shape. Prior to scanning, embryos were submersed in CystoCon Ray II (iothalamate meglumine) contrast agent for 1 h, and then scanned at 7.5-µm resolution. Neonates were scanned at 19-µm resolution without contrast agent to allow resolution of the bone. Scans were then reconstructed and landmarked using Meshlab (Version 1.3.2, Visual Computing Lab, meshlab.sourceforge.net) (embryos) or Amira (Version 5.2, FEI) (neonates). Landmarks for embryos were as developed by Percival et al.[50]. Neonate landmarks for the cranium are from Gonzalez et al.[71], with the addition of the landmarks on the mandible. Landmarks for each age group were placed by a single observer who was blinded to a genotype. A total 38 landmarks were placed on the embryos and 76 were placed on the neonates. Samples with shrinkage artifacts, or missing landmarks were removed from analysis.

**Geometric morphometrics**. Landmark data were imported into R, and Procrustes superimposition was performed to remove scaling and orientation differences between samples using the Geomorph package[72,73] in R[74]. Embryo data were regressed against tail somite number to remove ontogenetic effects before further analysis. Neonate data were regressed against centroid size only. Background effects due to lab of origin were mitigated by removing the difference between the wild-type groups from all specimens. A total of 187 neonates were analyzed and divided between groups as follows: WT (+/+) = 22, Flox/+ = 29, Neo/+ = 41, Flox/− = 10, ± = 25, Flox/+;Crect = 21, Flox/−;Crect = 19, Neo/Neo = 17, and Neo/− = 3 (w/all landmarks present). A total of 156 embryos were analyzed and divided between groups as follows: WT (+/+) = 27, Flox/+ = 15, Neo/+ = 30, Flox/− = 13, ± = 16, Flox/+;Crect = 16, Flox/−;Crect = 19, Neo/Neo = 12, and Neo/− = 8. Sample sizes were based on previous work[36,75,76]. Our power analysis shows that 10 embryos are needed to detect a 15–30% increase in variance and five embryos are needed to detect a 20–30% increase in variance with a power of 0.8. Due to the large number of genotypes, we focus on trends across the data set rather than between-group differences. The size- and lab-normalized shapes (Procrustes coordinates) were then regressed against Fgf8 level in Figs. 4 and 5. Residuals from both the age regression and the Fgf8 regression were obtained using a linear model, as implanted by the procD.Allometry function in geomorph. To represent these regressions as a single variable, we used the common allometric coefficient (CAC). When calculated from a pooled analysis with multiple groups, this is mathematically identical to a regression score[72] and plots these values as the dependent variables against the independent variables (Fgf8 level and tail somite stage).

**Modeling of phenotypic variance**. To model the relationship between Fgf8 expression and the phenotypic mean and within-genotype variance, we used Morrissey's model for the quantitative genetics of nonlinear G–P maps[34]. This model shows how the phenotypic mean is determined by the functional relationship between developmental processes ($\epsilon$) and the phenotype ($z$):

$$\bar{z} = \int f(\epsilon) N(\epsilon, \bar{\epsilon}, \sigma_\epsilon^2) \mathrm{d}\epsilon, \quad (1)$$

where $f(\epsilon)$ is the functional relationship between the developmental process and the phenotype and $N(\epsilon, \bar{\epsilon}, \sigma_\epsilon^2)$ is the normal distribution of developmental values (Fgf8 expression) with the specified mean and variance. The relationship between developmental and phenotypic variance is given by

$$\sigma_z^2 = \Phi^2 \sigma_\epsilon^2, \quad (2)$$

where

$$\Phi = \int f'(\epsilon) p(\epsilon) \mathrm{d}\epsilon, \quad (3)$$

$f'(\epsilon)$ is the first derivative of function $f(\epsilon)$ and $p(\epsilon)$ is the frequency of specific developmental values.

**Modeling of the G–P curve**. We fit the phenotype to Fgf8 expression at E10.5 and at P0 using a nonlinear least-squares regression to a von Bertalanffy curve of the formula:

$$z = L_m - (L_m - L_0) \mathrm{e}^{-k\epsilon}, \quad (4)$$

where $L_m$ is the maximum phenotype, $L_0$ is the mean phenotype at zero expression (y-intercept), and $k$ is a rate constant describing the decrease in slope per unit of $\epsilon$. In this curve, the initial rate of change of a phenotype given $\epsilon$ decreases at a rate proportional to $k$ until it reaches an asymptote ($L_m$).

**RNA collection for gene expression analyses**. E10.5 embryos were dissected into PBS on ice and snap frozen at −80 °C. Heads were dissected from between the mandibular arch and the hyoid arch. All RNA work was performed on the RNA

extracted from the head. RNA was extracted in batch preps using Trizol. cDNA was made from 500 ng of RNA in a 20-µl reaction mix using an iScript cDNA synthesis kit (Bio-Rad).

**qPCR**. Reverse transcription quantitative real-time PCR (RT-qPCR) was performed as previously described[77]. Briefly, we use a C1000 Thermal Cycler with a CFX96 Real-Time System (Bio-Rad). Forward and reverse primers, 2 µl of cDNA, RNase-free dH₂O, and SYBR-Select Master Mix (Thermo-Fisher), containing dNTPs, iTaq DNA polymerase, MgCl₂, SYBR Green I, enhancers, stabilizers, and fluorescein, were manually mixed in a 20-µl reaction to amplify each cDNA of interest. Primer sequences were GAPDH (F: 5′-AGGTCGGTGTGAACGGATTTG-3′; R: 5′-GGGGTCGTTGATGGCAACA-3′) and FGF8 (F: 5′-GTAGTTGTTCTCCAG-CACGAT-3′; R: 5′-GACAGGTCTCTACATCTGCAT-3′). Each sample was run in triplicate, all results were normalized to the expression of GAPDH, and fold changes were calculated using the delta–delta C(t) method[78]. Primers for qRT-PCR were selected for optimal G/C concentrations and tested for ideal melt curves and optimized for amplification efficiency: GAPDH, 92% at 61.5 °C and FGF8, 102% at 61.6 °C[79]. Primers for Fgf8 were located in the 3′ end of the transcript to prevent detection of nonfunctional transcript generated from the Neo or LacZ insertions. Real-time PCR quantification of the RNAseq data was performed as follows. cDNA was generated using the Maxima First Strand Kit (Thermo-Fisher) and amplified using the IDT mastermix and IDT PrimeTime probes and primers (Mm. PT.58.10694850, Mm.PT.58.7996582, Mm.PT.58.45983184, Mm.PT.58.29112396, Mm.PT.58.33292921, Mm.PT.58.43880967, Mm.PT.58.42634782.g, Mm. PT.58.33469229, and Mm.PT.58.41340681.gs). Samples were run on an Applied BioSystems QunatiStudio 6. Data were normalized by averaging Gapdh and β-actin expression levels. ddCT values were used in all downstream analysis. Correlation analysis was performed in R. The mean deltaCT for the controls was calculated before the log transformation for each sample, resulting in a slight alteration of the wild-type mean from 1.

**RNAseq**. RNA quality was assessed using an Agilent TapeStation and RIN scores of 9–10 were obtained. Stranded mRNA libraries for sequencing were prepared from ~1 µg of total RNA using the TruSeq Stranded mRNA library prep kit and Illumina protocol. The indexed libraries were quantitated for pooling by qPCR using a Kapa Library Quantification Kit and the pooled libraries were sequenced on two successive 75-bp high-output sequencing runs on an Illumina NextSeq 500 sequencer. An average of 46 million reads per sample was obtained. Reads were mapped using HT-Seq count, and then data were analyzed using DESeq2. Correlation analysis was run on the normalized counts, and other analyses were performed using the fold-change data. The gene lists used in the analysis are presented in Supplementary Table 3.

**Statistical note**. All P values are based on two-tailed tests unless otherwise noted.

**Code availability**. A code for all analysis as well as associated landmark data files can be found at http://www.ucalgary.ca/morpho/code-and-raw-data.

**Data availability**. RNAseq data have been uploaded to GEO with accession number GSE87366 and are available at https://www.ncbi.nlm.nih.gov/geo/query/acc.cgi?token=mtiveeyipxmhvkr&acc=GSE87366.
    Morphometric data are available with the analysis code at http://www.ucalgary.ca/morpho/code-and-raw-data. All raw data are available at the FaceBase Hub: (www.facebase.org) with accession number FB00000927: https://www.facebase.org/data/recordset/#1/isa:dataset./*::facets::N4IghgdgJiBcDaoDOB7ArgJwMYFM4gCoQAaEJHMbACznhADEAhABldYE4AmAdhAF0AvoKA@sort(release_date::desc::,id).

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

## Acknowledgements

This work was supported by grants NIH R01 2R01DE019638 to R.S.M. and B.H., NSERC 238992-17 to B.H. and C.C.R., and NIDCR R01 DE019843 to T.J.W. We thank Richard Hawkes for his valuable comments on the manuscript.

## Author contributions

R.M.G., J.L.F., B.H., R.S.M. and T.J.W. designed the experiments. R.M.G., J.L.F., I.C. and K.D. generated the embryos for analysis. R.M.G. and F.J.S. did the microCT scanning and landmarked the embryos. R.M.G. and B.H. analyzed the morphometric data. B.R. and K.D. generated the RNA, DNA, and ran the qPCR along with C.L.L., J.L.F. and R.M.G. analyzed the qPCR data. R.M.G., C.C.R. and P.G. analyzed the RNAseq data. C.C.R. analyzed the S.N.P. data. N.M.Y., J.M.C., C.C.R., R.S.M., B.H., J.L.F. and R.M.G. helped interpret the data and develop the initial model. J.M.C. generated the mathematical modeling. R.M.G., J.L.F. and B.H. wrote the paper. All authors revised and approved the final manuscript.
