## [Peer Review File · Nature Communications]

Reviewers' comments:

Reviewer #1 (Remarks to the Author):

CONTEXT AND GENERAL ASPECTS OF THE PAPER'S MESSAGES

In developmental systems and beyond, the target phenotype cannot be "too" sensitive to environmental or intrinsic noise, or even to some mutations. To make this statement scientific, it is necessary to say what "too" means. There is a vast evolutionary biology literature on robustness and canalisation that does so. For our purposes here, one may say that natural selection leads to architectures, genetic circuits, or regulatory processes which effectively buffer perturbations as long as the systems (developmental trajectories or other phenotypes) are close to their natural (normal) state. However if the system is strongly perturbed, those buffering forces have no reason to be effective (indeed there has been no evolutionary selection for buffering in that regime). This lack of buffering will lead to greater sensitivity to the perturbation (larger range of responses, steeper slopes in the response curves) and greater variability of the phenotype (across repetitions, genotypes etc).

To link all this to the work submitted in this manuscript, use these 2 identifications:

- the "perturbation" becomes the modification of dosage of the signaling factor Fgf8 in mice

- the "phenotype" becomes an index (a number) built from the many traits quantifying mouse embryonic facial morphology

Applying the previous general "theory" to a real system, the authors determine the response curve, that is the dependence of their index (the phenotype) as a function of the Fgf8 dosage (the perturbation). From these experimental measurements, they show that,

in line with the general canalisation picture, their response curve is flat near its natural operating point and it is steeper away from it. This is interpreted as release of (cryptic) variation as the buffering processes (against both environmental and genetic perturbations) break down. Were the response curve to be linear, no such canalisation would arise, explaining why the authors highlight the importance of the non-linear behavior of the response in justifying canalisation. Lastly, the authors consider variance of gene expression levels and of phenotype for a given genotype but their claims are muddled.

MAJOR POINTS

Some of these are real objections, some are requests for improvements as I found the authors did not make sufficiently clear the logic of several of their points.

(1) The author's claim -- that canalisation is understood via non-linearities rather than through buffering mechanisms -- does not satisfy me. Indeed what drives non-linearities of the response curve away from physiological conditions? Since these arise outside of the regime where natural selection operates, it seems difficult to accept the author's speculation that operation in the flat region and having steeper regions away from there are intrinsic to development.

(2) The authors stress that non linearities emerge from the dosage reduction of Fgf8, but never justify what is special about reduction vs increase. If you could increase

dosage, would you expect non linearities to appear too?

(3) As shown in figure 3, for low dosage of Fgf8 (less than 40%), one has the highlighted result of the paper. But I find unimpressive the increase of phenotypic variance. For instance, for P0, two of the mutants have a variance increase but the third mutant's variance is

reduced by about the same factor. If you did a shuffling amongst labels of mutants, you could

get a p-value. Would that p-value be less than 5%?

(3) The authors address the scenario whereby loss of robustness (below the 40% dosage value)

may be associated with increased gene expression variance (disruption). I had difficulty with

the logic of this part, starting with line 119. Specifically:

(a) although I agree that random dysregulation of individual gene expression will reduce correlations, it is dubious to consider that the genes will independently be randomized. Indeed, the randomization of one master gene will lead to strong correlations with its targets.

(b) for small perturbations, one expects each mean expression level to be linear in the change of Fgf8 dosage. Does that transpire from the data? Based on Figure 4 (perhaps you also wanted to indicate this in your text but the message didn't get through to me),

the variance within each mutant is not related to how far it is from the wild type. Thus if this is what you want to say (lower robustness is not associated with greater variance of expression within a genotype), do so explicitly.

(c) the claim that there is no evidence of dysregulation seems to be based on a lack of detected pattern for changes in the correlations. You seem to identify dysregulation and increased random variance. But when is variance just variance as opposed to random variance?

(4) Lines 138-152: the point (as I understand it) is Fgf8 dosage affects non linearities and not gene expression variances. But then how do you explain the larger phenotypic variances

at given genotype when one leaves the physiologically relevant regime?

(5) The paper may have been shortened too much because I guess from Figure 1 that although gene expression variance (even of down-stream targets) is not increased, this Figure suggests that there is an effect on the cell proliferation. Are you claiming that somehow proliferation leads to phenotypic variance for given genotype? I didn't find anything mentioning this in the text.

(6) The authors write: "We show that nonlinearity in the genotype-phenotype relationship for Fgf8 expression predicts phenotypic robustness.". That is not true, the fact that the relationship is non linear does not predict that the flat part of the curve coincides with the physiologically relevant region.

MINOR POINTS

(1) In their abstract, the authors say "Nonlinearities are a ubiquitous feature of development

that may link variation in development to phenotypic robustness." More specifically, what they

have in mind is a response curve that is flat in the biologically relevant range and steeper (non-linear) elsewhere. I'll take this as a definition of the robustness (otherwise the authors statements verge on tautologies) and then conclude that the contribution of this work is to show that in a specific experimental system, the response curve is as expected. I would thus recommend the authors put less stress on "theory" and the associated

(over-hyped) evolutionary biology terms.

(2) "Accordingly, such developmental relationships are a viable mechanistic explanation for canalization"... I wouldn't say it is a mechanistic explanation. At best, selection enforces insensitivity in the physiological regime but outside of that regime all bets are off.

(3) Fig 1: there are two parts, both are part B.

(4) The caption of Fig 2 doesn't mention the skulls part of that figure.

Reviewer #2 (Remarks to the Author):

Green et al. explore how phenotypic robustness changes upon gradual reduction gene function. Using quantitative morphometrics, they find that facial phenotypes only change in conditions that strongly reduce the Fgf8 mRNA dose in an established allelic series. They report that phenotypes are also more variable at these low doses, and that the expression of Fgf8-responsive genes is specifically reduced, while global correlations in gene expression do not change as a function of the specific Fgf8 allele. The authors conclude that the phenotypic response to altering Fgf8 mRNA expression is non-linear. This non-linearity explains why morphogenesis operates faithfully at different Fgf8 levels above a threshold level, and provides a mechanism for the accumulation of cryptic variation.

Many developmental biologists are well aware of the two central observations of the present manuscript: (1) Partial reduction in gene function, e.g. in heterozygotes, very often does not lead to strong phenotypes, and (2) the outcome of developmental processes is more variable in mutants than in the wild type (This has already noted by Waddington decades ago, as duly cited by the authors). However, very few studies so far have carefully quantified these observations, which is a prerequisite for elucidating the underlying mechanisms. By using their previously established morphometric analysis pipeline in an established allelic series of Fgf8 mutants the authors provide a valuable data set to work towards this.

While the quantification of the phenotypic output is one of the strong aspects of this paper, I have some problems with the quantification of the fgf8 mRNA levels in the different genotypes. The authors measure mRNA levels in whole embryos or neonates, but they do not address if the observed reduction results from lower mRNA levels in individual cells, a lower number of fgf8 mRNA producing cells, or a combination of the two. Furthermore, I fail to see statistical tests to examine between which of the allelic combinations there are statistically significant changes in fgf8 mRNA expression (data in Supplementary Figure S1C). It is therefore unclear to me if the data really represent a continuous series of nine different levels of gene function. It would also be of interest to assess more clearly for which

genotypes the phenotypic output shows statistically significant differences. This is an important point to address, as strong effects on phenotypic only appear to be observed in the three genotypes with two mutated alleles.

I also note that, in its current form, the paper only superficially addresses which molecular mechanisms underlie the observed non-linear relationship between genotype and phenotype. The authors demonstrate that FGF- responsive gene expression shows a similar non-linear response in the allelic series as the phenotypic output. It would be interesting to determine if gene expression is the first process that responds non-linearly to *fgf8* mRNA levels, or if these non-linearities can already be observed at the level of FGF protein production or signal transduction. Although this analysis would entail a new set of experiments, it would help to pinpoint at which level of biological organization the observed non-linearities arise, and thereby significantly increase our mechanistic understanding of the phenomena reported in the paper.

Additional points

Major:

- Figure 1B needs to be better explained in the main text and/or figure legend.
- The measurement of *Fgf8* expression levels is crucial for the interpretation of many results in the present paper. The authors need to perform a statistical analysis to evaluate if the measurements reported in Supplementary Figure S1C are significantly different between the genotypes.
- A more detailed description of the RNAseq experiments is required in the Methods section.

Minor:

- Supplementary Figure S1C: Two data series with different *Fgf8* expression levels are labeled as Flox+/CRECT, this needs to be corrected.
- The authors state in the methods section that their *Fgf8* primers have an amplification efficiency of 102%. It is hard for me to see how the efficiency in a qPCR experiment can be above 100%.

Signed: Christian Schröter

Reviewer #3 (Remarks to the Author):

This paper by Green et al. addresses an important question which is the underlying basis of phenotypic robustness. The authors generated transgenic lines to modulate the expression of *Fgf8*, a critical regulator of vertebrate development. They show that *Fgf8* expression (as

quantified by qRT-PCR) relates to facial morphology (quantified by geometric morphometrics) in a nonlinear manner. This nonlinearity translates to robustness of average facial shape and shape variance to variation in average Fgf8 levels within a range of 40%-100% WT Fgf8 expression levels (although the low dose is not quantified in great detail). This robustness is in contrast to substantial sensitivity of average shape and shape variance to Fgf8 expression below 40% of WT levels. The authors argue that phenotypic variance may stem from the shape of dose/response curve and presumably small differences in Fgf8 expression levels among individuals (the qPCR results suggest some stochasticity in Fgf8 expression). They go on to measure gene expression changes more globally in the Fgf8 allelic series via RNAseq. They find a similar nonlinear relationship between Fgf8 levels and downstream average fold change and no evidence for global gene expression variance among individuals. They conclude that multiple nonlinear genotype-phenotype curves embedded in developmental systems confer robustness.

This is an interesting study that showcases another example of nonlinearity in developmental genetics. Robustness has been mostly discussed in the past within the context of heat shock proteins partly due to historical reasons but also due to biases in the experimental design. Therefore, this study has the potential to be interesting to researchers in the wider field of biological robustness.

Major points:

- line 57 and rest of introduction: The authors discuss the link between nonlinearity and robustness as a novel hypothesis to test. However, Rendel's work (cited in the paper) already predicted such nonlinearities in genotype to phenotype mapping and these nonlinearities have been discussed previously in the context of biological robustness (for example, see a recent review by Felix and Barkoulas in Nature Rev Genetics PMID: 26184598 in which a conceptually similar nonlinearity between vulva development and egf genetic dose is described in *Caenorhabditis elegans*). I felt that the introduction to the biological question, at least as it currently appears in the manuscript, is rather incomplete.

- lines 62-63: It would also be good to have a little more background information on how Fgf8 affects face morphogenesis (developmental context). This would help the non-specialist audience in the field to appreciate the design of the allelic series and would also introduce aspects of the subsequent analysis of downstream gene expression (Fig 4).

- line 66-73: The current description of the allelic series in the main text and rationale of experimental design is unclear. The quantification of fgf8 expression level in the transgenic lines is very central to the story, so I would move panel S1C to the main figures. Do the CRECT lines add anything to the manuscript in the end? (I assume yes because these are the lines right at the border of the robustness range but this is not clearly discussed in the paper).

- line 92: I couldn't find where these supplemental data are located.

- line 107-118: I felt that the two "alternative" hypotheses to explain the variation in the

sensitivity to Fgf8 levels are also not written in a very clear way. This is partly because the authors do not go back to discuss Fig. 1B to show visually what the mean. It would be nice to include a cartoon describing the two possibilities. If there is nonlinearity in downstream responses to Fgf8 levels and stochasticity in fgf8 expression, wouldn't this lead to an increase in downstream gene expression variance at the population level ? The discussion of the two models can be improved.

- What happens to facial shape and variance upon Fgf8 overexpression ? Do the authors expect similar nonlinearities ? The authors only discuss decrease in Fgf8 expression in the manuscript.

- Why do the authors see the same trend with "all genes" as in "downstream targets" in Fig 4F vs H ? This is not discussed in the text.

- It would be interesting to see independent fgf8 dose/response curves for selected downstream genes.

Other comments:

- Fig. 1 Typo "A" should be in front of General models, not "B". The choice of the word "mechanism" for the X axis in panel A is unclear.

Panel B is not cited in the text and the figure legend is not very explanatory. For example, it is not obvious why variance of downstream genes is stable (bottom panel, second from left) ?

- Fig S1C: I believe one transgenic line is mislabelled on X (Flox/-, CRECT)

- Fig. 2 legend why panels B and D are presented with different color code ? Some panels are not discussed at all in the legend.

- Fig. S2 This figure would benefit by including arrows pointing to dysmorphic regions and some annotation (3 different skulls ? WT ? mutant ?)

- color code in Fig.3 makes data points hard to distinguish

- Some affiliations are missing underneath the title (5,6)

Guide to Revisions:

Overview: We thank the reviewers for their thorough and thoughtful comments. In response to these reviews, we have extensively revised the paper and added new data. We believe that these revisions have substantially improved the paper and are grateful for the constructive criticisms of the reviewers. Below, we detail the changes that we have made. Here is an overview of these changes:

- 1) Many comments, particularly those of reviewer 1, stem from the fact that the paper was originally formatted and written for *Nature* and the consequent limit on length and number of references did not allow us to sufficiently contextualize our study within the literature and explain the conceptual background. As detailed below, we actually agree with most of the points raised and have revised the paper to address them. This has resulted in a much more fulsome introduction and discussion. Both sections are effectively completely rewritten. In addition, we have reformatted the paper such that it is now divided into Introduction, Results, Discussion and Methods, as is usual for *Nature Communications*.
- 2) In response to several comments about the basis for the nonlinear curve and its relationship to phenotypic variance, we have now related our results quantitatively to Morrisey's formalization of non-linear genotype-phenotype maps. Importantly, our observed changes in variance closely resemble the predicted increase in variance after fitting a von Bertalanffy curve to the data and applying Morrisey's model.
- 3) We increased the sample sizes of embryos for morphometric analysis. Concerns were raised about statistical power. While some of these may stem from a misunderstanding (the pairwise comparisons among groups are less important than the estimation of the pattern of variation across groups), we do agree that increasing power is desirable. Our work to do this is the main reason why resubmission of the paper has been delayed to such an extent. It took many months to generate the required genotypes to increase our sample size due to unexpectedly poor breeding. The genotypes in the allelic series are difficult to work with and it has taken several years and very substantial effort to generate the unique sample that we have analyzed for this paper.
- 4) As requested, we have addressed all statistical issues and provided those results either in the supplemental data or within the paper itself.
- 5) As requested, we have added RT-PCR results to quantify both *Fgf8* and several downstream targets for E10.5 embryos. These data are consistent with the previous interpretation of the RNA-seq data and significantly add to the paper.

Below we have inserted specific explanations of revisions or replies to the concerns raised by reviewers.

Reviewer #1 (Remarks to the Author):

CONTEXT AND GENERAL ASPECTS OF THE PAPER'S MESSAGES

In developmental systems and beyond, the target phenotype cannot be "too" sensitive to environmental or intrinsic noise, or even to some mutations. To make this statement scientific, it is necessary to say what "too" means. There is a vast evolutionary biology literature on robustness and canalisation that does so. For our purposes here, one may say that natural

selection leads to architectures, genetic circuits, or regulatory processes which effectively buffer perturbations as long as the systems (developmental trajectories or other phenotypes) are close to their natural (normal) state. However if the system is strongly perturbed, those buffering forces have no reason to be effective (indeed there has been no evolutionary selection for buffering in that regime). This lack of buffering will lead to greater sensitivity to the perturbation (larger range of responses, steeper slopes in the response curves) and greater variability of the phenotype (across repetitions, genotypes etc).

Response: The reviewer is quite right. This issue stems from the overly brief text in the original paper. We have significantly expanded the introduction to address the issues raised here and this also comes up in the expanded discussion. The comment about a target phenotype being “too sensitive” relates to sloppy language on our part and that no longer occurs in the paper. We have added a much more fulsome treatment of the existing literature as we agree with the reviewer on this point.

To link all this to the work submitted in this manuscript, use these 2 identifications:

- the "perturbation" becomes the modification of dosage of the signaling factor Fgf8 in mice
- the "phenotype" becomes an index (a number) built from the many traits quantifying mouse embryonic facial morphology

Response: We agree and have added this language on page 4 in the second-last paragraph of the introduction.

Applying the previous general "theory" to a real system, the authors determine the response curve, that is the dependence of their index (the phenotype) as a function of the Fgf8 dosage (the perturbation). From these experimental measurements, they show that, in line with the general canalisation picture, their response curve is flat near its natural operating point and it is steeper away from it. This is interpreted as release of (cryptic) variation as the buffering processes (against both environmental and genetic perturbations) break down. Were the response curve to be linear, no such canalisation would arise, explaining why the authors highlight the importance of the non-linear behavior of the response in justifying canalisation. Lastly, the authors consider variance of gene expression levels and of phenotype for a given genotype but their claims are muddled.

Response: We have greatly revised both the introduction and the discussion to clarify our claims. We hope that our claims are no longer “muddled” and we agree with the reviewer that the previous version of the paper could have been clearer. It appears that clarity was lost in our attempt to fit the paper within the word limit for *Nature*.

MAJOR POINTS

Some of these are real objections, some are requests for improvements as I found the authors did not make sufficiently clear the logic of several of their points.

(1) The author's claim -- that canalisation is understood via non-linearities rather than through buffering mechanisms -- does not satisfy me. Indeed what drives non-linearities of the response

curve away from physiological conditions? Since these arise outside of the regime where natural selection operates, it seems difficult to accept the author's speculation that operation in the flat region and having steeper regions away from there are intrinsic to development.

Response: The reviewer raises an important point here. We address this in two ways. The first is that experimental demonstration of a nonlinear genotype phenotype map for a complex trait such as facial shape has not previously been attempted. Here, we use a genetic approach to generate nine gradations of the expression level of a key gene in craniofacial development. This is required to demonstrate the shape of the G-P map for this gene and is in and of itself significant. Further, we predict from general theory and then apply a quantitative genetic model to more specifically predict the changes in phenotypic variance along the curve. Our results are consistent with these predictions. This has not previously been done and so, in our view, fills in a significant gap in the literature. It is not clear to us why the reviewer finds this “unsatisfying.” Perhaps the issue is the obvious one that the mechanistic basis for the nonlinear GP map is not addressed by the work? Or perhaps the reviewer is arguing that the mechanistic basis for the nonlinear GP map could be understood as “buffering mechanisms.” Both of these issues are addressed in the responses below as well as in the revised discussion of the paper.

(2) The authors stress that non-linearities emerge from the dosage reduction of *Fgf8*, but never justify what is special about reduction vs increase. If you could increase dosage, would you expect non-linearities to appear too?

Response: This is an excellent point. We do not know whether an increase in *Fgf8* would result in a nonlinear GP map in the other direction. We agree with the reviewer that there is nothing inherently special about a perturbation that involves a decrease rather than an increase in expression. It is possible that excess *Fgf8* is always tolerated and it is also possible that variation in *Fgf8* expression produces no phenotypic change within some range around the Wt level but then produces a similarly nonlinear curve with increasing expression. This is a good question but one for another paper as it would require *de novo* generation of mouse genotypes. As it is, this paper already represents several years of work on these particular genotypes.

(3) As shown in figure 3, for low dosage of *Fgf8* (less than 40%), one has the highlighted result of the paper. But I find unimpressive the increase of phenotypic variance.

For instance, for P0, two of the mutants have a variance increase but the third mutant's variance is reduced by about the same factor. If you did a shuffling amongst labels of mutants, you could get a p-value. Would that p-value be less than 5%?

Response: We have increased the sample for morphometric analysis and have delayed resubmission of this paper by 9 months in order to include this expanded sample size. We do agree with the reviewer that more power is better here. That being said, the key issue is the relationship of the mean and variance across genotypes. For both the embryo and P0 samples, both are significantly different across genotypes. To address the reviewers

concern, however, we have also run the pairwise comparisons across groups. At P0, the low *Fgf8* expressing groups are significantly different from the higher expression groups (Flox/-;CRECT $p=0.029$ or lower, Neo/Neo $p=0.006$ or lower, Neo/- $p<0.001$). The Neo/- is not shown on the graph as we have a low sample size in that group due to the extreme phenotypic variability, though if we calculate it, it is an order of magnitude higher than for the other low expression groups (0.1 vs 0.02), which have a variance of approximately 1 order of magnitude higher than the high expressing groups (0.02 vs 0.004).

At E10.5, it is slightly more complicated as we do not see an increase in variance in the Flox/-;CRECT group. This is mentioned in the paper, but we believe this is due to the fact that *Fgf8* is lost in the ectoderm in these mice around E9.5, and so the effects of the deletion are just beginning. However, we see significant increases in the Neo/Neo and Neo/- groups. The Neo/Neo has roughly double the variance of WT (0.007 vs 0.015 $p=0.023$) and the Neo/- shows about a 3 fold increase from the Neo/Neo (0.042, $p<0.001$ for all groups). We hope that this addresses the reviewer's concern about statistical power.

(3) The authors address the scenario whereby loss of robustness (below the 40% dosage value) may be associated with increased gene expression variance (disruption). I had difficulty with the logic of this part, starting with line 119. Specifically:

(a) although I agree that random dysregulation of individual gene expression will reduce correlations, it is dubious to consider that the genes will independently be randomized.

Indeed, the randomization of one master gene will lead to strong correlations with its targets.

Response: The reviewer is quite right. One of the hypotheses that we test with the RNAseq data is that the increased phenotypic variance is related to increased variance of gene expression within genotypes. We test this indirectly via the pairwise correlations among transcriptomes within genotypes. We have revised the discussion of this test to make it clearer. We absolutely agree with the reviewers that genes are unlikely to be individually randomized. In fact, our results support this as we show strong nonlinear response in genes downstream to *Fgf8* in the low *Fgf8* genotypes. While this helps explain the larger phenotypic change in these genotypes, it does not explain the increased phenotypic variance within these genotypes. We hope that this is now significantly clearer in the revised paper.

(b) for small perturbations, one expects each mean expression level to be linear in the change of *Fgf8* dosage. Does that transpire from the data?

Response: *Fgf8* expression level does relate fairly linearly to *Fgf8* dosage. This is now shown in Figure 2. The point of the allelic series was to generate nine distinct, graded levels of *Fgf8* expression and this is, indeed, borne out by the data.

Based on Figure 4 (perhaps you also wanted to indicate this in your text but the message didn't get through to me), the variance within each mutant is not related to how far it is from the wild type. Thus if this is what you want to say (lower robustness is not associated with greater variance of expression within a genotype), do so explicitly.

Response. The variance of each genotype is actually quite clearly related to its position on the curve that describes the G-P map for *Fgf8* expression. This is confirmed by application of Morissey's model to our results. This is hopefully clearer in the revision.

(c) the claim that there is no evidence of dysregulation seems to be based on a lack of detected pattern for changes in the correlations. You seem to identify dysregulation and increased random variance. But when is variance just variance as opposed to random variance?

Response: Yes, the source of this confusion is probably our use of the term "dysregulation." In the literature, dysregulation is often used to refer simply to changes in expression as the result of some perturbation. We distinguish between changes in the mean expression level of downstream and variable response of downstream genes to modification of *Fgf8* level. By "dysregulation" we are referring to the latter kind of change. Arguably, changes in mean change expression, even if large, may actually be highly regulated. We have tried to make this clearer in the revision.

(4) Lines 138-152: the point (as I understand it) is *Fgf8* dosage affects non linearities and not gene expression variances. But then how do you explain the larger phenotypic variances at given genotype when one leaves the physiologically relevant regime?

Response: Yes, this is an interesting question! The finding that we do not show evidence of increased variance of gene expression in the genotypes with increased phenotypic variance is a central point of this paper. The nonlinear GP map argument implies that a given variance of gene expression will result in a different phenotypic variance depending on the location along the curve that describes that map. These results imply that the increased phenotypic variance is occurring either through many small changes in variance across many genes or at a level of mechanism above the gene expression level such as tissue level responses to the changes in gene expression. Fleshing this out in mechanistic terms is very difficult and represents a next phase in this work. This is beyond the scope of the current paper.

(5) The paper may have been shortened too much because I guess from Figure 1 that although gene expression variance (even of down-stream targets) is not increased, this Figure suggests that there is an effect on the cell proliferation. Are you claiming that somehow proliferation leads to phenotypic variance for given genotype? I didn't find anything mentioning this in the text.

Response: This issue flows from the previous one. Cell proliferation data is one of the lines of evidence that we are pursuing in the next phase of this work. The gene expression to protein translation relationship is another. As is always the case with developmental biology, every finding generates several more questions. Understanding the mechanistic basis for canalization is a tremendously difficult. Our results point towards view that the answer to that question does not lie so much in easy to understand mechanisms like heat shock proteins but rather in more complex, emergent properties of developmental systems. How this happens is not going to be answered within a single study and one has draw the line around a coherent body of experimental work somewhere. We have, after several years, chosen to do so with the present paper. If the reviewer will only accept a complete

mechanistic explanation of canalization, however, we think that may be asking too much of a single paper.

(6) The authors write: "We show that nonlinearity in the genotype-phenotype relationship for Fgf8 expression predicts phenotypic robustness.". That is not true, the fact that the relationship is non linear does not predict that the flat part of the curve coincides with the physiologically relevant region.

Response: The shape of the curve predicts the phenotypic variance along the curve. We have made this clearer in the text. That flows from the theory we present as well as from Morissey's model. Of course, the shape of the curve does not explain the shape of the curve. Again, this relates to the larger question of what the reviewer would consider here to be a sufficiently meaningful and significant insight into canalization to justify the study. We believe that this paper meets those criteria and would challenge the reviewer to identify another study that has already filled in the gap in knowledge that we address here.

MINOR POINTS

(1) In their abstract, the authors say "Nonlinearities are a ubiquitous feature of development that may link variation in development to phenotypic robustness." More specifically, what they have in mind is a response curve that is flat in the biologically relevant range and steeper (non-linear) elsewhere. I'll take this as a definition of the robustness (otherwise the authors statements verge on tautologies) and then conclude that the contribution of this work is to show that in a specific experimental system, the response curve is as expected. I would thus recommend the authors put less stress on "theory" and the associated (over-hyped) evolutionary biology terms.

Response: We beg to differ with the reviewer on this. It is not expected that the expression level of a key gene will exhibit a nonlinear G-P map. Another equally plausible possibility is that the expression levels of such genes are themselves buffered while systems remain highly sensitive to them. Much of the earlier literature on canalization and gene regulatory networks assumes this sort of dynamic where the aspects of the system stabilize key nodes within it. Further, it is commonly argued that complex traits exhibit mostly additive (linear) genetic variation because nonlinearities in developmental mechanism get averaged out by development. Here we show that development is actually quite insensitive to a large amount of variation in a key developmental gene while remaining highly sensitive to variation in the expression of that gene outside of a particular range. While nonlinear GP maps are not novel, this has not been shown before in a genetic model for a complex morphology. As such, it is highly relevant to understanding the evolvability of morphology. Further, the idea that genes that play key roles in development also contribute to variation in a trait has been used to privilege genes for genome wide association studies. Our results provide a reason to question this practice which has the potential to provide highly misleading results about the genetics of complex traits.

It is a tautology that our model generates changes in phenotypic variance only in that it is a mathematical necessity. That is a common property of quantitative models in biology. In this sense, all deterministic models are tautological. To a large extent, the entire field of quantitative genetics is tautological in this sense. Yet, that does not make those models any

less useful or the insights gained from the any less valid.

In terms of “over-hyped” terms in evolutionary biology, one of us (Hallgrimsson) has argued elsewhere for more precise definitions of terms such as integration and canalization precisely to avoid over hyping. We are not overhyping anything here. We have done an extensive and novel analysis of the genotype-phenotype map for a key gene in craniofacial development that shows how the shape of the GP map leads to modulation of phenotypic variance independently of genetic variance. We fail to see where we have overhyped our results here, especially as we have been very careful not to overinterpret them.

(2) "Accordingly, such developmental relationships are a viable mechanistic explanation for canalization" ... I wouldn't say it is a mechanistic explanation. At best, selection enforces insensitivity in the physiological regime but outside of that regime all bets are off.

Response: Again, what constitutes mechanism is a matter of perspective. This study is mechanistic in that we have generated variation in the expression of a key developmental gene via genetic models, predicted an outcome, modelled it quantitatively, and observed that outcome. That fits the definition of mechanisms in most of science. In developmental biology, “mechanism” often appears to refer to the latest range of molecular techniques rather than a particular type of logical explanation . There is always another level to pursue and every observation generates another question. In our case, we obtained relevant gene expression and transcriptomic data that tests and rejects a particular mechanistic explanation. This points towards a range of mechanism but does not provide a specific molecular, cellular or tissue level explanation for canalization.

(3) Fig 1: there are two parts, both are part B.

Corrected – thank you!

(4) The caption of Fig 2 doesn't mention the skulls part of that figure.

Corrected – thank you!

Reviewer #2 (Remarks to the Author):

Green et al. explore how phenotypic robustness changes upon gradual reduction gene function. Using quantitative morphometrics, they find that facial phenotypes only change in conditions that strongly reduce the Fgf8 mRNA dose in an established allelic series. They report that phenotypes are also more variable at these low doses, and that the expression of Fgf8-responsive genes is specifically reduced, while global correlations in gene expression do not change as a function of the specific Fgf8 allele. The authors conclude that the phenotypic response to altering Fgf8 mRNA expression is non-linear. This non-linearity explains why morphogenesis operates faithfully at different Fgf8 levels above a threshold level, and provides a mechanism for the accumulation of cryptic variation.

Many developmental biologists are well aware of the two central observations of the present manuscript: (1) Partial reduction in gene function, e.g. in heterozygotes, very often does not lead to strong phenotypes, and (2) the outcome of developmental processes is more variable in

mutants than in the wild type (This has already noted by Waddington decades ago, as duly cited by the authors). However, very few studies so far have carefully quantified these observations, which is a prerequisite for elucidating the underlying mechanisms. By using their previously established morphometric analysis pipeline in an established allelic series of Fgf8 mutants the authors provide a valuable data set to work towards this.

Response: We thank the reviewer for their thoughtful comments on our paper and its contribution to the literature.

While the quantification of the phenotypic output is one of the strong aspects of this paper, I have some problems with the quantification of the fgf8 mRNA levels in the different genotypes. **The authors measure mRNA levels in whole embryos or neonates, but they do not address if the observed reduction results from lower mRNA levels in individual cells, a lower number of fgf8 mRNA producing cells, or a combination of the two.**

Response: We did not perform single-cell RT-PCR in this study and so cannot answer the question of whether the number of Fgf8 expressing cells is affected by dosage or the amount produced per cell. This is an excellent question that will be pursued in future work. We have every reason to assume that the amount of mRNA is reduced within individual cells, but don't know that this is the case. It is possible to imagine other scenarios including interaction between cells and alterations in the anatomical shape of the Fgf8 expression domain. These are interesting questions and ones that may contain clues about the origin of the increased variance with low Fgf8 expression. However, this is beyond the scope of the current study.

Furthermore, I **fail to see statistical tests to examine between which of the allelic combinations there are statistically significant changes in fgf8 mRNA expression (data in Supplementary Figure S1C)**. It is therefore unclear to me if the data really represent a continuous series of nine different levels of gene function. It would also be of interest to assess more clearly for which genotypes the phenotypic output shows statistically significant differences. This is an important point to address, as strong effects on phenotypic only appear to be observed in the three genotypes with two mutated alleles.

Response: This is a good point. We have now included statistical tests for comparisons of Fgf8 levels across genotypes and this is reported in the supplementary data. The overall ANOVA across genotypes is highly significant at $p < 1 \times 10^{-7}$. This, alone, is sufficient basis on which to conclude that the genotypes vary in *Fgf8* level across the allelic series. Of course, in a graded series, some values are fairly similar (particularly the adjacent genotypes in the series) and so would require a large sample to get a significant p value in an adjusted pairwise test. Even so, many are significantly different after FDR adjustment for multiple comparisons as we now show in a table in supplementary data. In particular, the WT sample differs significantly from in 5 out of 8 pairwise comparisons. Given fairly small samples, this should more than satisfy this concern. Supplementary Figure 1 has now been moved Figure 2 in the main paper.

I also note that, in its current form, the paper only superficially addresses which molecular

mechanisms underlie the observed non-linear relationship between genotype and phenotype. The authors demonstrate that FGF- responsive gene expression shows a similar non-linear response in the allelic series as the phenotypic output. **It would be interesting to determine if gene expression is the first process that responds non-linearly to fgf8 mRNA levels, or if these non-linearities can already be observed at the level of FGF protein production or signal transduction.** Although this analysis would entail a new set of experiments, it would help to pinpoint at which level of biological organization the observed non-linearities arise, and thereby significantly increase our mechanistic understanding of the phenomena reported in the paper.

Response: These are excellent points. See comments in responses to reviewer 1's comments about the scope of the current paper. We agree that this is one of the next steps in this work, but we feel that this would really be a separate study. For this revision, we tried to obtain the data to add the protein level to this paper. However, we were not able to get sufficient numbers of embryos for all genotypes during the past 9 months as we were also producing samples for morphometrics and RNA. We have preliminary data on protein production but we are not sufficiently confident in it at this point to include it and at the rate that we are able to produce embryos, it would likely be another 6-9 months before we obtain sufficient data. If protein levels relate linearly to mRNA, then this pushes a mechanistic explanation for our results to a higher level of development which opens the study to the same kind of criticism but about some other factor (e.g. cellular response to *FGF8* protein production). There is no particularly compelling reason to believe that adding protein to the study would radically change the nature of the present study.

Additional points

Major:

- Figure 1B needs to be better explained in the main text and/or figure legend.

Response: We have revised both the figure and the figure legend for clarity.

- The measurement of Fgf8 expression levels is crucial for the interpretation of many results in the present paper. **The authors need to perform a statistical analysis to evaluate if the measurements reported in Supplementary Figure S1C are significantly different between the genotypes.**

Response: We have quantified both Fgf8 expression and that of key downstream genes using RT-PCR. This is the basis of a new figure (Figure 6) in the paper. We have also performed the statistical analysis required to verify that the differences among genotypes are significant and these are provided in supplemental table 1. We hope that this addresses the reviewers concern on this point.

- A more detailed description of the RNAseq experiments is required in the Methods section.

Response: We have provided a more detailed description of the RNAseq work in the Methods section as requested.

Minor:

- Supplementary Figure S1C: Two data series with different Fgf8 expression levels are labeled as Flox+/CRECT, this needs to be corrected.

Response: We have corrected this error. We thank the reviewer for noticing it!

- The authors state in the methods section that their Fgf8 primers have an amplification efficiency of 102%. It is hard for me to see how the efficiency in a qPCR experiment can be above 100%.

Response: You are correct in this, it cannot be. This is likely an artifact of a small amount of pipette error or a small amount of a PCR inhibitor present in the cDNA used for the dilution. The important part is that it is close to 100%.

Signed: Christian Schröter

Reviewer #3 (Remarks to the Author):

This paper by Green et al. addresses an important question which is the underlying basis of phenotypic robustness. The authors generated transgenic lines to modulate the expression of Fgf8, a critical regulator of vertebrate development. They show that Fgf8 expression (as quantified by qRT-PCR) relates to facial morphology (quantified by geometric morphometrics) in a nonlinear manner. This nonlinearity translates to robustness of average facial shape and shape variance to variation in average Fgf8 levels within a range of 40%- 100% WT Fgf8 expression levels (although the low dose is not quantified in great detail). This robustness is in contrast to substantial sensitivity of average shape and shape variance to Fgf8 expression below 40% of WT levels. The authors argue that phenotypic variance may stem from the shape of dose/response curve and presumably small differences in Fgf8 expression levels among individuals (the qPCR results suggest some stochasticity in Fgf8 expression).

They go on to measure gene expression changes more globally in the Fgf8 allelic series via RNAseq. They find a similar nonlinear relationship between Fgf8 levels and downstream average fold change and no evidence for global gene expression variance among individuals. They conclude that multiple nonlinear genotype-phenotype curves embedded in developmental systems confer robustness.

This is an interesting study that showcases another example of nonlinearity in developmental genetics. Robustness has been mostly discussed in the past within the context of heat shock proteins partly due to historical reasons but also due to biases in the experimental design. Therefore, this study has the potential to be interesting to researchers in the wider field of

biological robustness.

Response: We thank the reviewer for their thoughtful assessment of our paper.

Major points:

- line 57 and rest of introduction: The authors discuss the link between nonlinearity and robustness as a novel hypothesis to test. However, Rendel's work (cited in the paper) already predicted such nonlinearities in genotype to phenotype mapping and these nonlinearities have been discussed previously in the context of biological robustness (for example, see a recent review by Felix and Barkoulas in *Nature Rev Genetics* PMID: 26184598 in which a conceptually similar nonlinearity between vulva development and *egf* genetic dose is described in *Caenorhabditis elegans*). I felt that the introduction to the biological question, at least as it currently appears in the manuscript, is rather incomplete.

Response: We agree completely with the reviewer. The paper had been shortened, perhaps overly much, in order to fit to length for *Nature*. The revision for *Nature Communications* affords us the chance to better discuss the background literature and we have now done so. We thank the reviewer for drawing our attention to the Felix and Barkoulas paper. We have now cited that appropriately along with others who emphasize the importance of nonlinearity in explanations of robustness. The idea is older, of course, as Klingenberg and Nijhout made it in 1999 and Hallgrímsson et al did so in 2006. The contribution of this study is, as the reviewer notes, the quantification of the shape of the GP map and the fact that we relate this to phenotypic variance both formally through application of Morrisey's model and based on more general theory.

- lines 62-63: It would also be good to have a little more background information on how *Fgf8* affects face morphogenesis (developmental context). This would help the non-specialist audience in the field to appreciate the design of the allelic series and would also introduce aspects of the subsequent analysis of downstream gene expression (Fig 4).

Response: We agree completely and have added additional context into the introduction.

- line 66-73: The current description of the allelic series in the main text and rationale of experimental design is unclear. The quantification of *fgf8* expression level in the transgenic lines is very central to the story, so I would move panel S1C to the main figures. Do the CRECT lines add anything to the manuscript in the end ?(I assume yes because these are the lines right at the border of the robustness range but this is not clearly discussed in the paper).

Response: This is an excellent point. We have added a more fulsome and clearer explanation of why we chose these allelic series in order to generate the nine different *Fgf8* expression levels.

- line 92: I couldn't find where these supplemental data are located.

Response: This was due to an error in our original document that is now resolved. We apologize for this!

- line 107-118: I felt that the two “alternative” hypotheses to explain the variation in the sensitivity to Fgf8 levels are also not written in a very clear way. This is partly because the authors do not go back to discuss Fig. 1B to show visually what the mean. It would be nice to include a cartoon describing the two possibilities.

Response: We agree with the reviewer. We have clarified the conceptual background. The way we originally presented the two alternative hypotheses was a bit misleading as they are actually not mutually exclusive. This is explained more clearly now in the revised introduction. We have not added a cartoon to contrast these models as we think that the revised Figure 1 better captures the theoretical framework for the paper. We hope that our overhaul of the introduction better frames our work for the reviewer.

If there is nonlinearity in downstream responses to Fgf8 levels and stochasticity in fgf8 expression, wouldn't this lead to an increase in downstream gene expression variance at the population level ? The discussion of the two models can be improved.

Response: The reviewer is absolutely correct. We have better explained this now in both the introduction and in the discussion. This also relates to the use of the RNAseq data to test for changes in gene expression variance. We have explained this more clearly in the results section and hope that this allays the reviewer's concern on this point.

- What happens to facial shape and variance upon Fgf8 overexpression ? Do the authors expect similar nonlinearities ? The authors only discuss decrease in Fgf8 expression in the manuscript.

Response: We don't know and this is an excellent question. This is similar to a point raised by reviewer 1 that is addressed above.

- Why do the authors see the same trend with “all genes” as in “downstream targets” in Fig 4F vs H ? This is not discussed in the text.

Response: The trend is definitely weaker in “all genes” than in the downstream targets. This analysis is based on ordination of the transcriptomic data. This means that the change in PC1 are not necessarily driven by all genes even though the analysis is done using all genes. So, what this likely means is that the set of genes affected by Fgf8 expression is larger than the group we know to be downstream such that a signal is picked up when you combine all genes into an analysis. We don't know exactly which genes these are and although one could use the loadings to pick them out, the uncertainty in that one an individual gene basis would be high. We have added a sentence in the results section to explain this.

- It would be interesting to see independent fgf8 dose/response curves for selected downstream genes.

Response: This is an excellent suggestion. In response, we have performed RT-PCR for selected downstream genes and presented these results as Figure 7.

Other comments:

- Fig. 1 Typo “A” should be in front of General models, not “B”. The choice of the word “mechanism” for the X axis in panel A is unclear.
Panel B is not cited in the text and the figure legend is not very explanatory. For example, it is not obvious why variance of downstream genes is stable (bottom panel, second from left) ?

Response: In the interest of clarity we have changed this to “Dosage” and added several sentences describing each of these predictions.

- Fig S1C: I believe one transgenic line is mislabelled on X (Flox/-, CRECT)

Response: Fixed. Thank you!

- Fig. 2 legend why panels B and D are presented with different color code ? Some panels are not discussed at all in the legend.

Response: This has been updated

- Fig. S2 This figure would benefit by including arrows pointing to dysmorphic regions and some annotation (3 different skulls ? WT ? mutant ?)

Response: We have increased the annotation of these images.

- color code in Fig.3 makes data points hard to distinguish

Response: The colors have been changed.

- Some affiliations are missing underneath the title (5,6)

Response: This has been fixed, thank you!

Reviewers' comments:

Reviewer #1 (Remarks to the Author):

The authors have expanded significantly the experimental results and their analyses. The revised paper is clearer, more complete, and much more convincing.

With respect to the different points I had brought up in my first report, the authors have well addressed and responded to all major issues. For instance, in my point 3 bis, (sorry for my error in the numbering!), I had asked if phenotypic variance increased or not with the change in gene dosage; in the revision, the authors have cleared this up to my complete satisfaction, both via the experiments and by providing theoretical justifications. In fact, their use of Morissey's framework makes the paper much easier to read now.

Concerning the minor points listed in my previous report, I am satisfied also except perhaps for my first point. Indeed, to that point the authors respond as follows: "We beg to differ with the reviewer on this. It is not expected that the expression level of a key gene will exhibit a nonlinear G-P map." But in the introduction of the paper, they point to the opposite, saying in particular "Ligand-receptor binding, often described with a Hill function, is commonly nonlinear. The same is true for transcriptional regulation. Within tissues, processes such as the diffusion of a morphogen are nonlinear in ways that depend on anatomical context." It seems to me that the rebuttal and the paper are contradicting one another.

Since I am a constructive referee, let me point out some new minor points:

- In the Results section: "initial infinite rate" -> "initial rate"
- "given ϵ decreases at rate k " -> "given ϵ decreases at a rate proportional to k "
- In Fig 1B: GNR -> GRN (not defined)

Also, if the expression level is very low, it is difficult to believe one could have a constant variance (in any plausible scenario, the variance goes to 0 as the expression level goes to 0).

- In Fig 4: use same color codes in the top vs bottom sub-figures (blue and green have been exchanged).
- In the methods, please use a straight d in the d epsilon integrals.
- In the SM: Procustes -> Procrustes

Reviewer #2 (Remarks to the Author):

In the revised version of the manuscript "Developmental Nonlinearity Drives Phenotypic Robustness", the authors have significantly re-written the text, increased sample size, re-organized figures and added two new main figures (Fig. 4 and Fig. 7 in the revised manuscript). I find the revised version of the manuscript significantly easier to read, and most of my points brought up in the first round of reviews have been appropriately addressed. However, the new figures need to be significantly improved, and some further minor issues need to be addressed before publication is possible.

Major points:

The new figure 4 aims at a more quantitative description of the relationship between Fgf8 dosage and phenotype or variance in phenotype, while the new figure 7 shows correlations between Fgf8 dosage and downstream genes with the aim to pinpoint the molecular origins of the non-linearities observed earlier. Both approaches add significantly to the paper, however, in their current form the two figures are incomplete and/or confusing.

In Fig. 4 A,B, the authors seem to have chosen a subset of their data to fit the von Bertalaffny growth curve, but they do not indicate which data they chose and why. Panels C and D appears to be an interesting theoretical prediction, but I fail to see where this prediction is subsequently tested. E.g. in line 178/179 the authors state that "the predictions of the Morrissey model closely approximate our empirical results", without referring to a measurement.

These omissions in Fig. 4 make it difficult for me to understand how Fig. 5A, B goes conceptually beyond Fig. 4. How do the authors arrive at the conclusion that the model explains "54% of the phenotypic variance at E10.5 and 84% at P0" (Lines 181, 182)? It would be helpful here to have a definition and quantification of the amount of variance. I also do not understand why in Fig. 4 A,B, the authors plot "phenotypic value" on the y-axis, while in Fig. 5 A,B they plot "Regression residuals". This needs to be better explained. Figure 7 is confusing because it lacks a clear conclusion. Why do the authors choose to fit a polynomial to their data, instead of a different type of non-linear function? How do they evaluate if the non-linear function is more appropriate to explain the data?

Minor points

I still take an issue with the quantification of fgf8 mRNA levels by whole-embryo RT-PCR. This appears particularly problematic in the context of the CRECT-lines. I understand that these lines impair Fgf8 expression in the ectoderm, but not necessarily in mesodermal tissues such as the tailbud where Fgf8 mRNA is also expressed during the relevant developmental stages (e.g. Dubrulle & Pourquie, Nature, 2004). I therefore expect that whole-embryo RT-PCR will lead to an overestimation of Fgf8 mRNA levels in these genotypes. The authors need to at least comment on this issue.

The first two figures are now discussed in the introduction section. This could be moved to the results section.

There are a couple of typos throughout the text, e.g. line 124 "mechnisims"; line 202 "transcriptomic".

If the authors improved the manuscript on the points listed above, the paper will in principle be suitable for publication in Nature Communications.

Signed: Christian Schröter

Reviewer #3 (Remarks to the Author):

Developmental robustness is defined as the ability of an organism to resist either genetic or environmental fluctuation to produce an invariable phenotypic output. However, the underlying mechanisms that give rise to the robustness of development are not well understood. There are two dominating hypotheses in the field as to the mechanism by which this robustness is achieved: the first is that there exist distinct factors to buffer any variations, such as the heat shock proteins and other presumably unknown factors regulating variance, or secondly, that there are mechanisms within the developmental regulatory network itself such feedback and feed forward loops as to confer robustness to fluctuation. The two hypotheses are not mutually exclusive. In this paper, the authors seek to quantify the relationship between gene expression and phenotypic output in order to address the latter hypothesis for robustness. They propose that a nonlinear phenotypic response to changes in gene expression would confer substantial developmental robustness within a range of gene expression levels, and would also explain the large variation of phenotypes that arise in animals expressing under a certain threshold. The authors use an allelic series of the growth factor Fgf8 to generate a range of expression levels up to and including the wild-type level, and quantify face shape changes at two developmental time points to measure phenotypic output. They show that mean phenotype becomes sensitive to small differences in Fgf8 expression when its expression falls below 40 % of the wild-type level. They find that the shape variance increases when the Fgf8 expression is below 40 % of wild-type levels. Using RNA-seq, they did not find broad dysregulation of gene expression among individuals in low Fgf8 condition (with increased phenotypic variance) but they found that downstream genes may respond nonlinearly to Fgf8 expression.

The revised version of the manuscript includes some additional experiments (see Fig.7) / data analysis (including extra stats) and a mathematical model of phenotypic variance (see Fig. 4) that collectively add to the paper. The main strength remains the quantification of the gene to phenotype map for the developmental factor Fgf8 and its relationship to Morrisey's model and more general theory (which is added in this revised version). The main limitation remains that the study is based on a fairly limited allelic series with few data points (and without including perturbations above the wild-type levels). It also uses whole-organism approaches to quantify the perturbations and their molecular effects therefore losing in mechanistic (tissue specific) resolution. Another major drawback is the writing, which remains marred by typos/ mistakes/omissions substantially decreasing clarity. Some examples are listed below.

The authors have provided a more extensive introduction but some parts still remain difficult to follow, especially for the non specialised audience. For example, starting from lines 48-49, the relevance of Waddington's experiment is difficult to understand without providing any context. Another example is p2. line 56 where developmental stability is not defined at all.

The authors introduce Fgf8 and the hypotheses they are testing but this is way too abrupt. It would be helpful to introduce the gene and discuss its role in development and then provide some justification for using this particular gene in this study and explain the rationale of their hypothesis. (line 93 - 98 would fit better in the figure 1 legend).

Fig. 1A mechanism does not sound as the right term to use for the X axis. The colours mentioned in the legend do not match.

Fig. 1B warrants some explanation in the figure legend. The title says "FGF pathway and canalization" but this does not seem very helpful to the reader. Also, the variance panels can be removed or at least better explained in the main text or legend. It is not true that variation in gene expression is constant for any genotype - the data in Figure 2 C shows there is much higher variation in expression in the heterozygotes and the hypomorph allele than the wt or closer to null animals.

line 93: The authors hypothesize that the Fgf8 expression relates linearly to genotype (Fig. 1Bi). However, there is no confirmation or rejection of this hypothesis using their Fgf8 expression data (Consider getting rid of Fig. 1Bi).

line 122: The authors claim that they have chosen the allelic series because of nine different gradations of Fgf8 dosage, however, the statistical tests in supplemental table 1 show that there are fewer significantly different dosage conditions.

p7, line 193: "...The only exception is for E10.5 Fgf8;Crect embryos." The authors do not discuss how they interpret this result

line 209 "Genes known to be downstream.....expression below 40%" Is this significant though ? Still good to explain why a similar trend is seen in "all genes" as provided in the response to reviewers.

line 223: It is difficult to argue that the changes described here are biologically relevant to any real "compensation"

line 233: "These changes occur most markedly in the genotypes expressing 40% or less Fgf8 compared to the wild-type"

This would be a good validation result but I am not sure I see this in the figure.

lin 238 - it would be good to add a final sentence to conclude how these results relate to the previous analysis of downstream genes in Fig 6.

Fig. 7 How is it possible that Fgf8 expression values go up to 2 ? Also why do hets express more fgf8 than the Wt ?

More comments:

p1 line 36-38: The authors may want to modify the following sentence to make clearer. "...across this series has a non-linear...". It is not clear what "this series" is.

line 50: not just in fruit flies and mice

line 86: "...quantitative genetic theory to formally relate..."

p3 line 97: "...between Fgf8 expression and morphology.." Need to specify craniofacial morphology.

lines 99 - 106: Would be nice if the authors discuss this study at the end. It is unclear why the authors are using Fgf8 rather than Sonic hedgehog and is interesting how their results relate to their previous studies.

line 116 - 117: "...of the neomycin insertion. The second series, Fgf8;Crect, uses..."

line 130 - 142: Would be nice to have a summary of the findings from the paper in this paragraph, as well as the predictions.

p7 line 202: "...transcriptomic PC1" is a new sentence

Line 220 and Fig 6: Explanation of how the MapK pathway etc relates to Fgf8 and facial morphogenesis is unclear

p10 line 280: "...low Fgf8 levels." Fgf8 needs to be italicized.

p11 line 283: Remove the hyphen between gene and regulatory.

p14 line 412: The sample sizes do not add up to 187

lines 733 -734: "Note the thin layer of blue present over the entire embryo" - suggesting what?

line 751: "...P< 0.001..."

line 759: "...z = 0.01765...". Same in line 226.

line 736: "...between 3-22 samples per group." There are only two spots for the Flox/-;CRECT data set on the graph.

Figure 1Bii: Do the authors mean to abbreviate gene regulatory network so should it be "GRN" and not "GNR" ?

Fig. 2C: The legend on the right is redundant to the x-axis.

Fig 2. Presumably the homozygous null allele is lethal, but it would be nice to state why it was not included

Fig. 3 There is no comment on the discrepancy between the data from UCSF and UMass for the Neo allele of Fgf8 in B - is there a statistical difference, and what is the explanation proposed? Also, text in the right part below PC1 should be P0 I suppose

Fig 4 it would be nice to swap the colour of the blue and green lines to match the data in A and B. It would be helpful to the reader to keep colours consistent between Fig. 3 and Fig. 4 for relative Fgf8 levels. For example, red colour denotes low and high Fgf8 levels in Fig. 3 and Fig. 4 levels.

Fig. 6A: There is no SE interval for non-linear relationship of Rictor with Fgf8.

Fig. 5 and 6: It would be helpful to increase the tick marks on x-axis, to see clearly the 0.4 value which is the threshold for Fgf8 expression below which the phenotype becomes sensitive.

Fig 6: I believe the asterisks representing the significance of differences at low levels of Fgf8 expression are misplaced.

Sup Table 1: Please be consistent with the allele names - this is important for the non specialised audience. Wild-type alleles are denoted by "WT" and "+" sometimes and Flox or CRECT appear written in many different ways.

Fig S1 It would help to have a wild-type embryo and arrows pointing to the features mentioned in the legend to make them accessible to a general audience.

Consider correcting DeSeq2 to DESeq2, RNAseq instead of RNA-seq, RT-PCR instead of rt-PCR.

Consider being more consistent in the format and information included in the figure legends.

Response to reviewers:

We thank the reviewers for their detailed and constructive critiques, and we believe that their comments continue to improve the paper. Our responses to the reviewers are presented here in bold and changes to the text are made in red.

Reviewer #1 (Remarks to the Author):

The authors have expanded significantly the experimental results and their analyses. The revised paper is clearer, more complete, and much more convincing.

With respect to the different points I had brought up in my first report, the authors have well addressed and responded to all major issues. For instance, in my point 3 bis, (sorry for my error in the numbering!), I had asked if phenotypic variance increased or not with the change in gene dosage; in the revision, the authors have cleared this up to my complete satisfaction, both via the experiments and by providing theoretical justifications. In fact, their use of Morissey's framework makes the paper much easier to read now.

Response: Thank you!

Concerning the minor points listed in my previous report, I am satisfied also except perhaps for my first point. Indeed, to that point the authors respond as follows: "We beg to differ with the reviewer on this. It is not expected that the expression level of a key gene will exhibit a nonlinear G-P map." But in the introduction of the paper, they point to the opposite, saying in particular "Ligand-receptor binding, often described with a Hill function, is commonly nonlinear. The same is true for transcriptional regulation. Within tissues, processes such as the diffusion of a morphogen are nonlinear in ways that depend on anatomical context." It seems to me that the rebuttal and the paper are contradicting one another.

Response: Yes, in re-reading our text and the rebuttal, we can see how one might see this as a contradiction. What we are trying to say there is that although many component processes such as receptor-ligand relationships are known to be nonlinear, a G-P map is the result of many such processes acting at different levels and might, therefore, be fairly linear. In fact, this assumption is often implicit in many approaches to complex trait genetics (e.g. most GWAS methods) that are designed to detect additive variation only. We should have been more explicit about this in our response and also explained this better in the paper. We have done so now (lines 78-82). As an aside, our groups have tested the assumption that genetic variation for craniofacial shape is mostly additive in other work and this shows a surprising amount of non-additive variation. I think we are basically in agreement with the reviewer on this and appreciate his comment but we recognize that we did not explain this properly.

Since I am a constructive referee, let me point out some new minor points:

- In the Results section: "initial infinite rate" -> "initial rate" –

"given ϵ decreases at rate k " -> "given ϵ decreases at a rate proportional to k "

Response: Thanks, please note these sentences have been moved to the methods section

- In Fig 1B: GNR -> GRN (not defined)

Response: This has been fixed and defined.

Also, if the expression level is very low, it is difficult to believe one could have a constant variance (in any plausible scenario, the variance goes to 0 as the expression level goes to 0).

Response: This is an excellent point that we did not discuss in the paper but should have. As the reviewer perceives, if the variance of gene expression goes down with the mean expression level, then this changes the expected change in phenotypic variance along the curve. The reviewer is correct that at 0, the variance would also be 0 and so one might suspect that at 0.2, the variance would begin its approach to 0 (at least in so far as it would start to approach the hard limit of 0). However, we have actually tested for heteroscedasticity in the Fgf8 expression data. Levene's test for homogeneity of variance across groups (ANOVA on the absolute mean deviations) shows that there is no difference in variance across the genotypes. F value = 1.41 P=0.204. We have added this to lines 150-151 and thank the reviewer for pointing this out.

! In Fig 4: use same color codes in the top vs bottom sub-figures (blue and green have been exchanged).

Response: Thank you! This has been changed

- In the methods, please use a straight d in the d epsilon integrals.

Response: We have done this.

- In the SM: Procustes -> Procrustes

Response: We have done this.

We thank the reviewer once again for an excellent critique of this work.

Reviewer #2 (Remarks to the Author):

In the revised version of the manuscript "Developmental Nonlinearity Drives Phenotypic Robustness", the authors have significantly re-written the text, increased sample size, re-organized figures and added two new main figures (Fig. 4 and Fig. 7 in the revised manuscript). I find the revised version of the manuscript significantly easier to read, and most of my points brought up in the first round of reviews have been appropriately addressed. However, the new figures need to be significantly improved, and some further minor issues need to be addressed before publication is possible.

Major points:

The new figure 4 aims at a more quantitative description of the relationship between Fgf8 dosage and phenotype or variance in phenotype, while the new figure 7 shows correlations between Fgf8 dosage and downstream genes with the aim to pinpoint the

molecular origins of the non-linearities observed earlier. Both approaches add significantly to the paper, however, in their current form the two figures are incomplete and/or confusing.

In Fig. 4 A,B, the authors seem to have chosen a subset of their data to fit the von Bertalanffy growth curve, but they do not indicate which data they chose and why. Panels C and D appears to be an interesting theoretical prediction, but I fail to see where this prediction is subsequently tested. E.g. in line 178/179 the authors state that “the predictions of the Morrissey model closely approximate our empirical results”, without referring to a measurement.

Response: We realize that the discussions of the Von Bertalanffy growth curve and of Morrissey’s model are a bit mixed together, which makes it difficult to understand. We are addressing this by making substantial revision to lines 160-173 as well as to the areas mentioned. Essentially, figure 4 is a conceptual model that establishes a hypothesis which is explained by figure 5. The Morrissey model was generated using the means and variances of the Fgf8 expression data to generate predictions for the variances in the phenotypic data.

These omissions in Fig. 4 make it difficult for me to understand how Fig. 5A, B goes conceptually beyond Fig. 4. How do the authors arrive at the conclusion that the model explains “54% of the phenotypic variance at E10.5 and 84% at P0” (Lines 181, 182)? It would be helpful here to have a definition and quantification of the amount of variance. I also do not understand why in Fig. 4 A,B, the authors plot “phenotypic value” on the y-axis, while in Fig. 5 A,B they plot “Regression residuals”. This needs to be better explained.

Response: We have clarified this in the text. The von Bertalanffy model (figure 4) generates predictions that are tested in figure 5. Since these are predictions, and not actual values, we thought it made sense to show that in the axis, we have commented on this in the text. The data explained represents the proportion of variance explained by fitting the data to the von Bertalanffy curve.

Figure 7 is confusing because it lacks a clear conclusion. Why do the authors choose to fit a polynomial to their data, instead of a different type of non-linear function? How do they evaluate if the non-linear function is more appropriate to explain the data?

Response: We agree that there were some issues with Figure 7 as originally presented. We have repeated the PCR to use primers in Exons 5/6 to be consistent with the data presented in the rest of the paper. We have also chosen to just show the linear curves as, in the revised data, there was little difference between the linear and non-linear models. We have added several sentences to give a clearer conclusion to this figure at the end of the manuscript.

Minor points

I still take an issue with the quantification of fgf8 mRNA levels by whole-embryo RT-PCR. This appears particularly problematic in the context of the CRECT-lines. I understand that these lines impair Fgf8 expression in the ectoderm, but not necessarily in mesodermal tissues such as the tailbud where Fgf8 mRNA is also expressed during the relevant developmental stages (e.g. Dubrulle & Pourquie, Nature, 2004). I therefore

expect that whole-embryo RT-PCR will lead to an overestimation of Fgf8 mRNA levels in these genotypes. The authors need to at least comment on this issue.

Response: The RT-PCR is done on only on the heads, not on the whole embryo. We failed to make this clear in the methods. The following sentence has been added. "Heads were dissected from between the mandibular arch and the hyoid arch. All RNA work was performed on the RNA extracted from the head." We have also made a note of this in the text. We apologize for this omission.

The first two figures are now discussed in the introduction section. This could be moved to the results section.

Response: We set this up this way as we feel that figure 1 is essentially a visual abstract and set-up of the non-linear model, however we agree with the reviewer that the modified version of Figure 2 should in fact be in the results section and have moved it accordingly.

There are a couple of typos throughout the text, e.g. line 124 "mechnisims"; line 202 "transcriptomic".

Thank you.

If the authors improved the manuscript on the points listed above, the paper will in principle be suitable for publication in Nature Communications.

Signed: Christian Schröter

Reviewer #3 (Remarks to the Author):

Developmental robustness is defined as the ability of an organism to resist either genetic or environmental fluctuation to produce an invariable phenotypic output. However, the underlying mechanisms that give rise to the robustness of development are not well understood. There are two dominating hypotheses in the field as to the mechanism by which this robustness is achieved: the first is that there exist distinct factors to buffer any variations, such as the heat shock proteins and other presumably unknown factors regulating variance, or secondly, that there are mechanisms within the developmental regulatory network itself such feedback and feed forward loops as to confer robustness to fluctuation. The two hypotheses are not mutually exclusive. In this paper, the authors seek to quantify the relationship between gene expression and phenotypic output in order to address the latter hypothesis for robustness. They propose that a nonlinear phenotypic response to changes in gene expression would confer substantial developmental robustness within a range of gene expression levels, and would also explain the large variation of phenotypes that arise in animals expressing under a certain threshold. The authors use an allelic series of the growth factor Fgf8 to generate a range of expression levels up to and including the wild-type level, and quantify face shape changes at two developmental time points to measure phenotypic output. They show that mean phenotype becomes sensitive to small differences in Fgf8 expression when its expression falls below 40 % of the wild-type level. They find that the shape variance increases when the Fgf8 expression is below 40 % of wild-type levels. Using RNA-seq, they did not find broad dysregulation of gene expression among individuals in low Fgf8 condition (with increased phenotypic variance) but they found that downstream genes

may respond nonlinearly to Fgf8 expression.

The revised version of the manuscript includes some additional experiments (see Fig.7) / data analysis (including extra stats) and a mathematical model of phenotypic variance (see Fig. 4) that collectively add to the paper. The main strength remains the quantification of the gene to phenotype map for the developmental factor Fgf8 and its relationship to Morrisey's model and more general theory (which is added in this revised version). *The main limitation remains that the study is based on a fairly limited allelic series with few data points (and without including perturbations above the wild-type levels)*. It also uses whole-organism approaches to quantify the perturbations and their molecular effects therefore losing in mechanistic (tissue specific) resolution. Another major drawback is the writing, which remains marred by typos/ mistakes/omissions substantially decreasing clarity. Some examples are listed below.

The authors have provided a more extensive introduction but some parts still remain difficult to follow, especially for the non specialized audience. For example, starting from lines 48-49, the relevance of Waddington's experiment is difficult to understand without providing any context. Another example is p2. line 56 where developmental stability is not defined at all.

Response: We agree and have revised the first paragraph to provide the appropriate context with greater clarity. The first two paragraphs are completely rewritten and we hope that this meets with the reviewers approval. The paper does not deal with developmental stability and so we have simply removed the reference to this. The issue came up in the previous versions because our use of "robustness" as opposed to "canalization". Robustness is a more general concept that includes both canalization and developmental stability. We used this term in the title because the term "canalization" may not be as widely understood. It would take significant space to define these terms in a nuanced and scholarly way. This would be more appropriate in a review (we have one in preparation currently). For the purpose of this paper it is only necessary to contextualize canalization within robustness as we have now done in the revised version.

The authors introduce Fgf8 and the hypotheses they are testing but this is way too abrupt. It would be helpful to introduce the gene and discuss its role in development and then provide some justification for using this particular gene in this study and explain the rationale of their hypothesis. (line 93 - 98 would fit better in the figure 1 legend).

Response: We had added a paragraph in the last version on the role of Fgf8, but it came after the original statements of the hypothesis. This has been reorganized. We agree with the reviewer and have removed lines 93-98 from where they were in the text, parts have been added to the figure legend, and parts discussed at the end of the introduction.

Fig. 1A mechanism does not sound as the right term to use for the X axis. The colours mentioned in the legend do not match.

Response: We agree that the red was more brown than red and have made it "redder". We have also expanded the legend to make it easier to understand.

Fig. 1B warrants some explanation in the figure legend. The title says "FGF pathway and

canalization” but this does not seem very helpful to the reader. Also, the variance panels can be removed or at least better explained in the main text or legend. It is not true that variation in gene expression is constant for any genotype - the data in Figure 2 C shows there is much higher variation in expression in the heterozygotes and the hypomorph allele than the wt or closer to null animals.

line 93: The authors hypothesize that the Fgf8 expression relates linearly to genotype (Fig. 1Bi). However, there is no confirmation or rejection of this hypothesis using their Fgf8 expression data (Consider getting rid of Fig. 1Bi).

Response: To address these two comments, as well as a comment from Reviewer 1, we added the results of a Levene’s test for the gene expression in the area of the figure 2 results. The Levene’s test result is non-significant and therefore we fail to reject the hypothesis that gene expression variance is constant across groups. We have added additional discussion to the legend for figure 1B, saying that these are predictions.

line 122: The authors claim that they have chosen the allelic series because of nine different gradations of Fgf8 dosage, however, the statistical tests in supplemental table 1 show that there are fewer significantly different dosage conditions.

Response: We have downplayed this language slightly by saying nine alleles generating a gradation of Fgf8 dosage. In order to show that each group is fully statistically distinct from all other groups would require a massive increase in sample size. We never claim that all groups are statistically independent from one another, only that all groups represent a gradation of gene expression. All of our hypotheses relate to a gradation and understanding changes along this gradation, not to being able to separate changes in adjacent groups.

p7, line 193: “...The only exception is for E10.5 Fgf8;Crect embryos.” The authors do not discuss how they interpret this result

Response: Good point. We have added the following “likely due to the fact that CRECT activates at E10.5. By P0, these show a significant increase.”

line 209 “Genes known to be downstream.....expression below 40%” Is this significant though ? Still good to explain why a similar trend is seen in “all genes” as provided in the response to reviewers.

Response: Yes, the genotypes with Fgf8 expression below 40% are significantly different from the others for all groups. However, the heterozygous genotypes are not significantly different from each other for the Fgf8 downstream target genes. We have revised the paragraph to clarify this.

line 223: It is difficult to argue that the changes described here are biologically relevant to any real “compensation”

line 233: “These changes occur most markedly in the genotypes expressing 40% or less Fgf8 compared to the wild-type” This would be a good validation result but I am not sure I see this in the figure.

Response: Compensation is a hypothesis that is predicted by our results, but not directly tested in this paper. We state that it “shows evidence for”, which we believe is justified by both the identification of genes with a negative correlation, and also by the identification of related Fgfs that are highly correlated. We have noted 1 result that looks promising from Figure 7, where there is a significant increase in transcript in one of the Neo groups compared to wildtype, which strongly implies compensation. The only way will know for sure that any of these genes compensate would be to overexpress them or knock them out in combination with Fgf8 which is out of the scope of this paper.

lin 238 - it would be good to add a final sentence to conclude how these results relate to the previous analysis of downstream genes in Fig 6.

Response: Thank you. We agree and have added one.

Fig. 7 How is it possible that Fgf8 expression values go up to 2 ? Also why do hets express more fgf8 than the Wt?

Response: This is a valid point and was a failure in the way we had performed this experiment. We had used a commercial designed assay for this experiment (and only this experiment) without paying attention to where in *Fgf8* it was located. In retrospect, we realized it was on exons 1 and 2, upstream of the deletion. It is likely, therefore, that additional, non-functional transcript was being generated. We have re-done this with a probe and primer pair located in exons 5 and 6 as was used in the rest of the paper.

More comments:

p1 line 36–38: The authors may want to modify the following sentence to make clearer. “...across this series has a non-linear....”. It is not clear what “this series” is.

Response: The abstract has been revised to fit the Nature Communications format. We hope we have improved its clarity in the process.

line 50: not just in fruit flies and mice

Response: The first paragraph has been greatly revised and this sentence rewritten.

line 86: “...quantitative genetic theory to formally relate...”

Thank you.

p3 line 97: “..between Fgf8 expression and morphology..” Need to specify craniofacial morphology.

Response: Thank you, we have tried to make sure that it is clear that we are always referring to craniofacial morphology.

lines 99 - 106: Would be nice if the authors discuss this study at the end. It is unclear

why the authors are using Fgf8 rather than Sonic hedgehog and is interesting how their results relate to their previous studies.

Response: Our decision to move away from Shh to Fgf signaling for this study was primarily a practical decision, as a good allelic series for loss of Shh is not available. We have added a sentence to the discussion on the previous paper. As explained in the introduction, it was important to conduct this experiment using a genetic model. In the direct treatment model that we used in the original Shh study, one cannot exclude the contribution of experimental (treatment dose) variation to among-individual variance. We have added a paragraph to the discussion to explain this (lines 306-311). We hope this meets with the reviewer's approval.

line 116 - 117: "...of the neomycin insertion. The second series, Fgf8;Crect, uses..."

Response: Check

line 130 - 142: Would be nice to have a summary of the findings from the paper in this paragraph, as well as the predictions.

Response: You, and the Nature Communications formatting checklist are in agreement here, it has been done.

p7 line 202: "...transcriptomic PC1" is a new sentence

Line 220 and Fig 6: Explanation of how the MapK pathway etc relates to Fgf8 and facial morphogenesis is unclear

Response: This paragraph has been revised.

p10 line 280: ".....low Fgf8 levels." Fgf8 needs to be italicized.

Response: Check

p11 line 283: Remove the hyphen between gene and regulatory.

Response: Check

p14 line 412: The sample sizes do not add up to 187

Response: Apologies, there was an error in one of the genotype sample sizes, it should be 10, not 8.

lines 733 -734: "Note the thin layer of blue present over the entire embryo" - suggesting what?

We have modified this sentence to say: Note the thin layer of blue present over the entire embryo showing the ectodermal CRE expression showing the ectodermal CRE expression

line 751: "...P< 0.001..."

line 759: "...z = 0.01765...". Same in line 226.

line 736: "...between 3-22 samples per group." There are only two spots for the Flox/-;CRECT data set on the graph.

Response: Oops, this has been corrected.

Figure 1Bii: Do the authors mean to abbreviate gene regulatory network so should it be "GRN" and not "GNR" ? **Yes, thank you.**

Fig. 2C: The legend on the right is redundant to the x-axis.

Fig 2. Presumably the homozygous null allele is lethal, but it would be nice to state why it was not included. **Added to the figure legend.**

Fig. 3 There is no comment on the discrepancy between the data from UCSF and UMass for the Neo allele of Fgf8 in B - is there a statistical difference, and what is the explanation proposed? Also, text in the right part below PC1 should be P0 I suppose

Response: There is a subtle difference in the groups, that was removed before the generation of this graph by removing the difference between wildtype groups (Methods – geometric morphometrics). We believe this is due to some amount of genetic drift within the colony, as well as changes in the fixation method between the labs. We agree though that this graph is confusing and now distracts from its original purpose of showing the subtle differences between the two series. We have returned it to two colors.

Fig 4 it would be nice to swap the colour of the blue and green lines to match the data in A and B. It would be helpful to the reader to keep colours consistent between Fig. 3 and Fig. 4 for relative Fgf8 levels. For example, red colour denotes low and high Fgf8 levels in Fig. 3 and Fig. 4 levels.

Fig. 6A: There is no SE interval for non-linear relationship of Rictor with Fgf8.

Fig. 5 and 6: It would be helpful to increase the tick marks on x-axis, to see clearly the 0.4 value which is the threshold for Fgf8 expression below which the phenotype becomes sensitive.

Fig 6: I believe the asterisks representing the significance of differences at low levels of Fgf8 expression are misplaced.

Response: You are correct in that two asterisks are missing, these have been replaced.

Sup Table 1: Please be consistent with the allele names – this is important for the non specialised audience. Wild-type alleles are denoted by "WT" and "+" sometimes and Flox or CRECT appear written in many different ways.

Fig S1 It would help to have a wild-type embryo and arrows pointing to the features mentioned in the legend to make them accessible to a general audience.

Consider correcting DeSeq2 to DESeq2, RNAseq instead of RNA-seq, RT-PCR instead of rt-PCR.

Consider being more consistent in the format and information included in the figure legends.

Response: Thank you. We have attended to these inconsistencies.

Reviewers' comments:

Reviewer #1 (Remarks to the Author):

The authors have satisfactorily responded to all my questions.

Reviewer #2 (Remarks to the Author):

In this revised version of their manuscript, Green et al. have made changes to the text and some figures that overall improve the work. My concerns with the quantification of fgf8 mRNA expression and the display and interpretation of the new figure 7 have been appropriately addressed.

Unfortunately, I still believe that figures 4 and 5 are still not appropriately described in the text. In other words, the authors have not addressed my question from the previous round of review as to where the measures for "phenotypic value" in Fig. 4 A,B derive from. At the moment line 167 of the text says "we fit the data..." without specifying what "the data" are. Are these values from the author's study? Then they need to specify how they have been calculated. How do they get from their multivariate measurements to the Common Allometric Component of shape - this needs to be explained, or an appropriate reference needs to be cited. The authors also have to specify why only four datapoints are shown. Finally, I would also like to see an estimate of the error in the respective panels. With respect to Fig. 5, the authors also have not explained how "regression residuals" in panels A, B were calculated.

Minor points

Line 112 "...and between genotypes" is a trivial statement.

Lines 114 – 117: This sentence appears to be redundant or needs to be explained better.

Lines 117 – 119: The way this is written is confusing. It sounds as if Fig. 1B showed the authors data on gene expression changes, but this is not the case. (These results are only presented in Fig. 6.) Please rewrite.

Line 141: Typo "is"

Line 155/156: Does this statement refer to Supplemental table 2?

Line 209: How can the PC plot in Fig. 6 explain variation in facial shape? The way I understand this PC plot is that PC1 captures 44% of the gene expression variance.

Fig. 7: The authors need to specify how they have normalized their qPCR data. I notice that the mean value for wt samples is slightly off 1.0;1.0 for many panels.

Signed: CS

Reviewer #3 (Remarks to the Author):

The authors have satisfactorily addressed/ responded to all points. This is a well-executed study on robustness and I support publication.

Response to reviewers 3:

Reviewer #1 (Remarks to the Author):

The authors have satisfactorily responded to all my questions.

Reviewer #2 (Remarks to the Author):

In this revised version of their manuscript, Green et al. have made changes to the text and some figures that overall improve the work. My concerns with the quantification of fgf8 mRNA expression and the display and interpretation of the new figure 7 have been appropriately addressed.

Response:

Thank you.

Unfortunately, I still believe that figures 4 and 5 are still not appropriately described in the text. In other words, the authors have not addressed my question from the previous round of review as to where the measures for “phenotypic value” in Fig. 4 A,B derive from. At the moment line 167 of the text says “we fit the data...” without specifying what “the data” are. Are these values from the author’s study? Then they need to specify how they have been calculated. How do they get from their multivariate measurements to the Common Allometric Component of shape - this needs to be explained, or an appropriate reference needs to be cited. The authors also have to specify why only four datapoints are shown. Finally, I would also like to see an estimate of the error in the respective panels.

Response:

We have reorganized the sentence to define the data we are using to fit the model earlier in the sentence and separated the details of the model build into a second sentence. We have also added a citation for the CAC score. It now reads: **To generate the curve used to test Morrissey’s model, we fit the *Fgf8* gene expression data, and the phenotypic data (3D landmark data) to a von Bertalanffy curve using least-squares regression. The phenotype data used was the regression score from a multivariate regression of our normalized Procrustes coordinates on *Fgf8* level – which generates single variable shape score⁴⁸.**

Its not that only 4 data points are shown, the curve represents all the data. We have highlighted 4 gene expression levels and their respective locations on the curves as these are levels we have modeled in panels C and D. A note to this effect has been added to the figure legend. **Colored dots highlight location on curves of 4 gene expression values that are modeled in B-C.** There is no error in this panel, as these are not true data points, just the resulting curves from the models. We show how these models fit to our data in figure 5.

We agree with the reviewer that some of the discussion of this figure was confusing as written as some of the text better fit into the following paragraph. This change has been made (and the resulting text highlighted in red).

With respect to Fig. 5, the authors also have not explained how “regression residuals” in panels A, B were calculated.

Response:

The “regression residuals” used in Figure 5 are the same regression residuals used in Figure 4. We have added the following to the methods section: **The size and lab normalized shapes (Procrustes coordinates) were then regressed against *Fgf8* level in figures 4 and 5. Residuals from both the age regression and the *Fgf8* regression were obtained using a linear model as implemented by the `procD.Allometry` function in `geomorph`. To represent these regression as a single variable, we used the common allometric coefficient (CAC). When calculated from a pooled analysis with multiple groups, this is mathematically identical to a regression score⁷² and plot these values as the dependent variables against the independent variables (*Fgf8* level and tail somite stage).**

Minor points

Line 112 “...and between genotypes” is a trivial statement.

Response:

I’m not sure we agree here. It is important to see how the gene expression variance changes between genotypes. We would argue that is one of the main purposes of the paper.

Lines 114 – 117: This sentence appears to be redundant or needs to be explained better
Lines 117 – 119: The way this is written is confusing. It sounds as if Fig. 1B showed the authors data on gene expression changes, but this is not the case. (These results are only presented in Fig. 6.) Please rewrite.

Response:

We agree this section was somewhat confusing and revised lines 114-119, this section now reads:

At the transcriptome level, we predict that there will be both compensatory and downstream gene expression changes (Figure 1B). We show that once *Fgf8* falls below a threshold level, there is both a change in the mean cranial shape and an increase in the variance of that shape. We further show that changes in phenotypic variance do not relate to increases in gene expression variance and that there are both non-linear and linear downstream gene expression changes.

Line 141: Typo “is”

Response:

Thank you.

Line 155/156: Does this statement refer to Supplemental table 2?

Response:

No it does not, supplemental table 2 is referred to in line 191

Line 209: How can the PC plot in Fig. 6 explain variation in facial shape? The way I understand this PC plot is that PC1 captures 44% of the gene expression variance.

Response:

You are correct, it should read gene expression.

Fig. 7: The authors need to specify how they have normalized their qPCR data. I notice that the mean value for wt samples is slightly off 1.0;1.0 for many panels.

Response:

We have added the following sentence to the methods: **The mean deltaCT for the controls were calculated before the log transformation for each sample resulting in a slight alteration of the wildtype mean from 1.**

Signed: CS

Reviewer #3 (Remarks to the Author):

The authors have satisfactorily addressed/ responded to all points. This is a well-executed study on robustness and I support publication.